# Impact of glacial meltwater on phytoplankton biomass along the Western Antarctic Peninsula
B. Jack Pan [1] ✉, Michelle M. Gierach[1], Sharon Stammerjohn [2], Oscar Schofield [3], Michael P. Meredith [4], Rick A. Reynolds [5], Maria Vernet [6], F. Alexander Haumann [7,8], Alexander J. Orona[9] & Charles E. Miller [1]

The Western Antarctic Peninsula is undergoing rapid environmental change. Regional warming is causing increased glacial meltwater discharge, but the ecological impact of this meltwater over large spatiotemporal scales is not well understood. Here, we leverage 20 years of remote sensing data, reanalysis products, and field observations to assess the effects of sea surface glacial meltwater on phytoplankton biomass and highlight its importance as a key environmental driver for this region's productive ecosystem. We find a strong correlation between meltwater and phytoplankton chlorophyll-a across multiple time scales and datasets. We attribute this relationship to nutrient fertilization by glacial meltwater, with potential additional contribution from surface ocean stabilization associated with sea-ice presence. While high phytoplankton biomass typically follows prolonged winter sea-ice seasons and depends on the interplay between light and nutrient limitation, our results indicate that the positive effects of increased glacial meltwater on phytoplankton communities likely mitigate the negative impact of sea-ice loss in this region in recent years. Our findings underscore the critical need to consider glacial meltwater as a key ecological driver in polar coastal ecosystems.

The ocean along the Western Antarctic Peninsula (WAP, Fig. 1) is a biological "hotspot" with estimated pelagic net primary production averaging ~1000 mg C m$^{-2}$ d$^{-1}$ nearshore (<100 km from the coast) and ~100 mg C m$^{-2}$ d$^{-1}$ offshore (>100 km from the coast) over the summer[1,2]. The productive phytoplankton community fuels the ecosystem in this region[3–5]. The WAP is also experiencing rapid environmental change—it is one of the fastest warming regions on Earth[6,7]. Mean warming recorded along the WAP reached 3.7 ± 1.6 °C during the twentieth century[6,8], more than six times the estimated 0.6 ± 0.2 °C global mean surface warming during this same period[9]. This warming trend persists into the twenty-first century[10–12], exemplified by a recent anomalously warm period in February 2020 (mean temperature anomaly of +4.5 °C relative to 1950–2019 summer baseline conditions)[13]. As a result, glacial melt rates along the Peninsula are accelerating as evidenced by both remote sensing measurements and numerical simulations[10,14,15]. While the long-term physical impact of glacial meltwater

on sea level has been extensively studied[16,17], its immediate biological effect and its function as an ecological driver are highly uncertain.

Most prior WAP marine ecological studies have focused on sea-ice dynamics due to its well-documented impact on phytoplankton biomass and seasonal succession[2,4,18], especially diatoms, a dominant phytoplankton group along the WAP[19,20]. Sea-ice not only serves as a physical barrier to wind-driven mixing but also hosts sea-ice algae in its brine channels. Sea-ice also offers a habitat beneath its underside, where algae can thrive, and it plays a critical role in the life cycle of Antarctic krill (*Euphausia superba*)[21]. These algae can enable "seeding" by providing an early-season diatom population that is released into the water column during ice melt and serving as a food source for Antarctic kill[18,19,22]. Additionally, sea-ice can accumulate micronutrients, such as iron, that are subsequently made available to phytoplankton upon melting[23]. During winter, the sea-ice cover limits wind-induced deep mixing[24,25] and

[1]Jet Propulsion Laboratory, California Institute of Technology, Pasadena, CA, USA. [2]Institute of Arctic and Alpine Research, University of Colorado Boulder, Boulder, CO, USA. [3]Department of Marine and Coastal Sciences, Rutgers University, New Brunswick, NJ, USA. [4]British Antarctic Survey, Cambridge, UK. [5]Marine Physical Laboratory, Scripps Institution of Oceanography, University of California San Diego, La Jolla, CA, USA. [6]Integrative Oceanography Division, Scripps Institution of Oceanography, University of California San Diego, La Jolla, CA, USA. [7]Helmholtz Centre for Polar and Marine Research, Alfred Wegener Institute, Bremerhaven, Germany. [8]Ludwig-Maximilians-Universität München, Munich, Germany. [9]Data Science & Artificial Intelligence Group, Ocean Motion Technologies, Inc., San Diego, CA, USA. ✉e-mail: oceanography@oceanmotion.tech

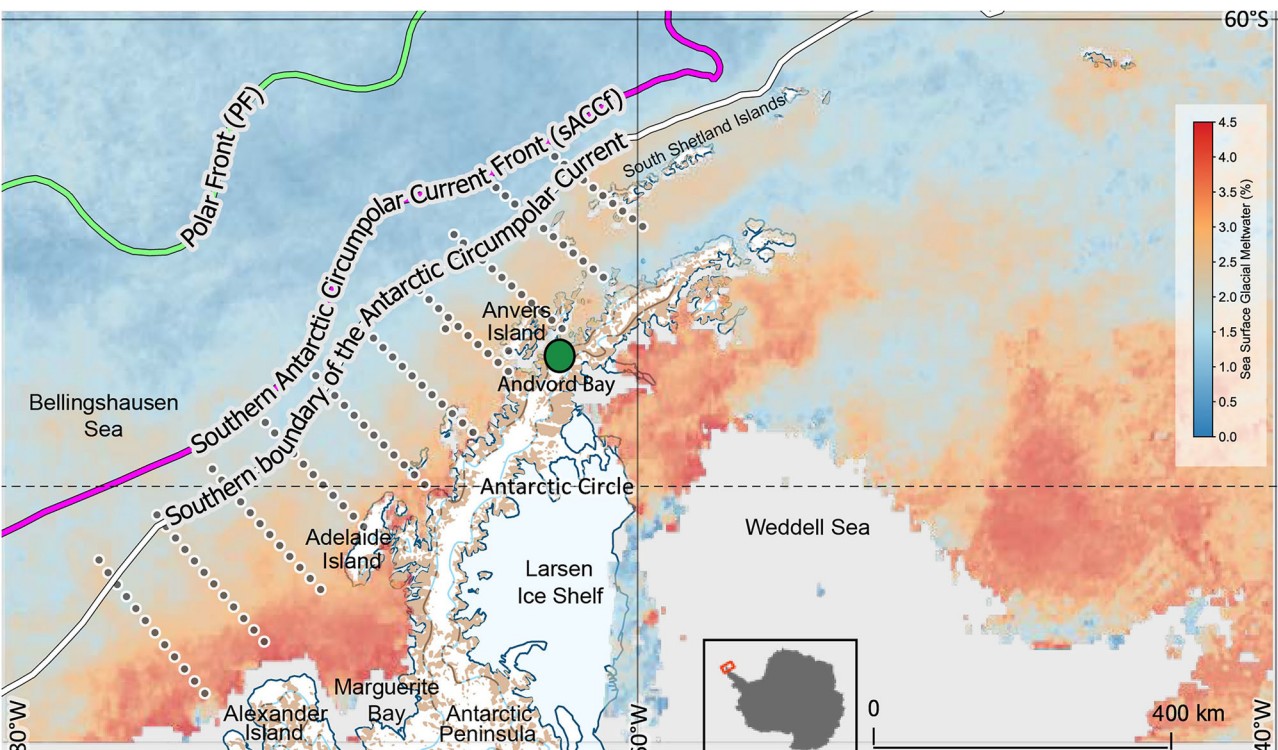

**Fig. 1 | Map of the Western Antarctic Peninsula showing satellite-derived sea surface glacial meltwater (sGMW) fraction averaged between September 2021 and February 2022.** Gray points represent the Palmer Long-Term Ecological Research Program's sampling grid. The green point identifies Andvord Bay, adjacent to Anvers Island. The locations of major ocean currents and fronts illustrate the co-location of the outer extent of sGMW with the southern boundary of the Antarctic Circumpolar Current and the shelf break in this region.

creates upper ocean stabilization for sustaining phytoplankton stock in the water column[26].

Recently, sea surface glacial meltwater (sGMW) has been identified as an additional environmental driver for phytoplankton productivity in the WAP region[19,27,28]. While prior work documented taxonomic shifts under varying meltwater regimes on a seasonal scale (e.g., shifts between cryptophytes and diatoms), our focus here is on sGMW's long-term influence on overall chlorophyll-a (chl-a) concentration, rather than individual functional groups. Glaciers along the WAP exhibit a range of terminus types—most are marine-terminating tidewater glaciers, but some are land-terminating. This distinction profoundly affects how meltwater enters the ocean[15,29]. Marine-terminating glaciers can undergo both submarine melting (where relatively warm circumpolar deep water erodes the glacial front at depth) and subglacial discharge (which often fosters localized upwelling of nutrient-rich deep water into the photic zone)[30]. Surface melting, iceberg calving, and glacial runoff also contribute varying amounts of freshwater and sediments[31,32]. These processes collectively influence the biogeochemistry and ecology of the coastal ocean, as meltwater can introduce micro- and macro-nutrients or alter light regimes—both of which are crucial drivers for phytoplankton growth[33]. Glacial meltwater can entrain both dissolved and particulate matter, including sediments, which may affect water clarity and nutrient dynamics. While overall sediment production in the WAP has historically been tempered by relatively low surface melt, some locales (such as Potter Cove) illustrate that glacial retreat can result in abundant sediment discharge[34,35]. These sediments may negatively impact light availability in nearshore waters. At the same time, recent studies found that glacially derived particles are enriched in iron and manganese, enhancing phytoplankton growth when these nutrients are bioavailable[36]. For example, in Andvord Bay (Fig. 1), a glacio-marine fjord in the northern WAP region, glacial meltwater has been found to positively impact total phytoplankton biomass, alter community composition, and influence its seasonal community succession between cryptophytes and diatoms[28].

Dissolved iron (dFe) concentration in this fjord ranged from 1 to 13 nM in late spring and 2 to 9 nM in fall, while particulate iron concentration, spanning both seasons, ranged from 100 to 1000 nM[36]. The spatial extent of sediment plumes in the offshore region remains poorly constrained; while strong winds or currents can disperse turbid waters beyond the immediate fjord, sediment-driven light attenuation is generally expected to be less pronounced offshore, where mixing dilutes the glacial plumes[35,37].

As glacial melt intensifies in the future, WAP sediment discharge could become a more consequential factor for both coastal and shelf phytoplankton ecology. Given the importance of primary producers in the WAP region, it is critical to understand glacial meltwater's impact over ecologically-relevant spatial and temporal scales[2,38]. Prior studies on this subject have focused on short-term in-situ observations[28] and the methodology development of an ocean-color-based sGMW product[39]. Here, we build on these approaches, leveraging observations from field measurements as well as multiple satellite remote sensing platforms and reanalysis products, covering a nearly 20-year period from 2002 to 2022 to demonstrate that, in addition to sea-ice dynamics, sGMW is an important environmental driver for phytoplankton ecology across the WAP. Our findings indicate that, in recent decades, the ecological impact of sea-ice loss on phytoplankton in this region is mitigated by a substantial increase in sGMW.

## Results and discussion

### Glacial meltwater drives regional phytoplankton growth

There is a significant correlation between sGMW fraction and chl-a concentration ($r = 0.70$, $p < 0.0001$, $n = 3860$), both in fjords and over the continental shelf of the WAP, independently showing consistency between in-situ and remotely sensed observations (Fig. 2a). This positive correlation is attributed to glacial meltwater discharge being a source of nutrients, particularly dFe, as well as being a proxy for buoyant meltwater-driven and wind-driven upwelling of deep nitrate and iron along the glacio-marine

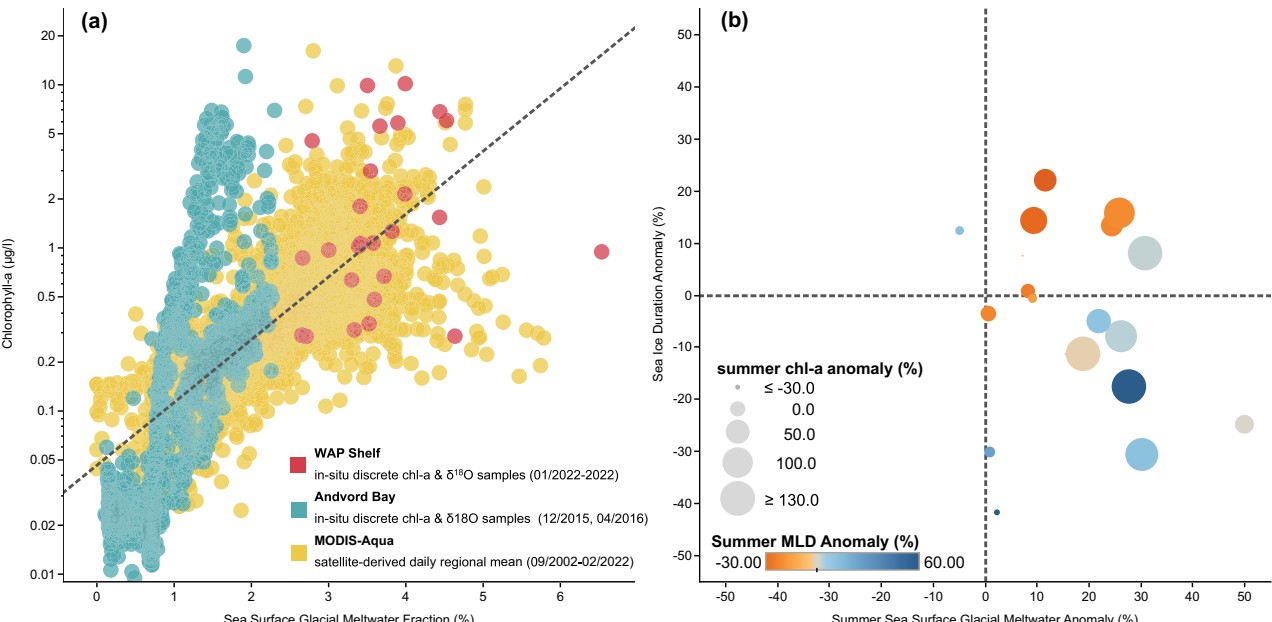

**Fig. 2 | Sea surface glacial meltwater and phytoplankton chl-a. a** Turquoise and red data points are based on in-situ observations in Andvord Bay (December 2015, April 2016) and over the continental shelf of the Western Antarctic Peninsula (January 2002–2022), respectively. Glacial meltwater fraction is derived from stable oxygen isotope and salinity measurements, and chlorophyll-a (chl-a) concentration from fluorescence measurements. Yellow data points are daily regionally averaged values derived from MODIS-Aqua level 3 remote sensing reflectance (September 2002–February 2022); regression line for all data points indicates $r = 0.7$, $p < 0.0001$, $n = 3860$. **b** Regionally averaged summer anomalies (D/J/F) by year based on 2002–2022 climatology, where the size of the data points represents summer chl-a concentration anomaly derived from MODIS-Aqua, and the color represents summer mixed layer depth anomaly based on field measurements conducted by the Palmer LTER Program. See "Methods" for more detail.

interface[36,40,41]. Moreover, sGMW has been found to induce surface layer stabilization and reduce the depth of mixing[24,42] (Fig. 2b upper right quadrant), thus leading to more optimal light conditions for phytoplankton growth[43,44]. It is important to note that higher surface chl-a in a shallower mixed layer depth (MLD) does not always indicate higher depth-integrated primary production or biomass. In particular, if the mixed layer is rapidly shoaled by freshening or sea-ice melt, the surface layer may exhibit temporarily high chl-a without necessarily reflecting a corresponding increase in total depth-integrated biomass. Nonetheless, the WAP's hydrography often allows surface chl-a to be an effective first-order indicator of phytoplankton biomass[1,18], especially when examining regional-scale processes over climatological and multi-decadal periods.

By examining these observations on an annual timescale, we track how sGMW affects water-column processes and phytoplankton biomass—both throughout the growing season and from year to year, thereby enhancing our understanding of its ecological role. Although we use chl-a as a proxy for phytoplankton biomass, we note that it may not always capture changes in species composition or depth-integrated biomass. Over an annual timescale, we find a general association between sGMW and chl-a (Fig. 2b right quadrants, Methods), but the relationship with MLD is more nuanced. While sGMW can freshen and stabilize the surface layer, sea-ice duration and wind forcing often exert a stronger influence on the seasonal-mean MLD. Hence, the co-occurrence of high sGMW and deeper (or shallower) MLD does not alone imply causation. Sea-ice duration can influence phytoplankton biomass in multiple ways. A longer ice season sometimes facilitates the accumulation of sea-ice algae that may seed phytoplankton blooms; it can also enhance post-melt stabilization and fertilization, potentially leading to higher chl-a in summer. However, shorter sea-ice duration can alleviate light limitation[45] earlier in the season, which can also result in high chl-a under favorable wind and nutrient conditions. Consequently, the relationship between sea-ice and summer chl-a can vary significantly among different years (Fig. 2). This result is consistent with prior studies in the WAP region that demonstrated sea-ice dynamics can impact phytoplankton biomass[1,46]. High chl-a concentrations during summer are

also observed after anomalously low sea-ice durations (Fig. 2b, lower right quadrant). This relation indicates that sea-ice duration, especially during the proceeding winter, is not the only key ecological driver for summer phytoplankton in the WAP region. Our results indicate that, even with low sea-ice duration anomalies, high phytoplankton biomass during the growing season can still be achieved in the presence of high sGMW content (Fig. 2). These results present a synoptic view of a coherent, region-wide ecological impact of sGMW on phytoplankton biomass across the entire WAP region and over a 20-year period.

MLD strongly influences phytoplankton in coastal polar regions—a shallower MLD enhances light availability, whereas a deeper MLD may increase nutrient input but can limit phytoplankton growth via light limitation and dilution. In the WAP, sea-ice, winds, and glacial meltwater shape MLD variability, with major impacts on phytoplankton biomass and distribution[19]. Summer sGMW content is also associated with both shallower (Fig. 2b upper right quadrant) and deeper MLD (Fig. 2b lower right quadrant). At first, the latter appears to contradict conventional understanding of sGMW's impact on surface ocean, where the presence of buoyant, fresh meltwater can enhance stratification leading to a shallower MLD[38,42]. While freshening by glacial meltwater is commonly expected to stabilize the upper ocean, our results show that some summers with high sGMW fraction also exhibit deeper mixed layers. This discrepancy most likely arises from transient or seasonal wind-driven mixing events, as well as interannual variability in sea-ice duration[24,30,47]. Consequently, the stabilizing impact of sGMW is not always reflected in the seasonal-mean MLD, even though sGMW remains an important driver of upper-ocean stratification. Moreover, prior in-situ observations of MLD were conducted with vessel-based instrument casts, hence they offer an instantaneous "snapshot" of the water column. However, when observations are averaged over the growing season (December–February), sGMW can also coincide with deeper MLD likely due to wind forcing[30,48]. Prolonged inshore observations have found that wind events, particularly katabatic winds, are an important mechanism for transporting glacial meltwater away from fjords' glacio-marine interfaces and towards offshore onto the wider continental shelf[49].

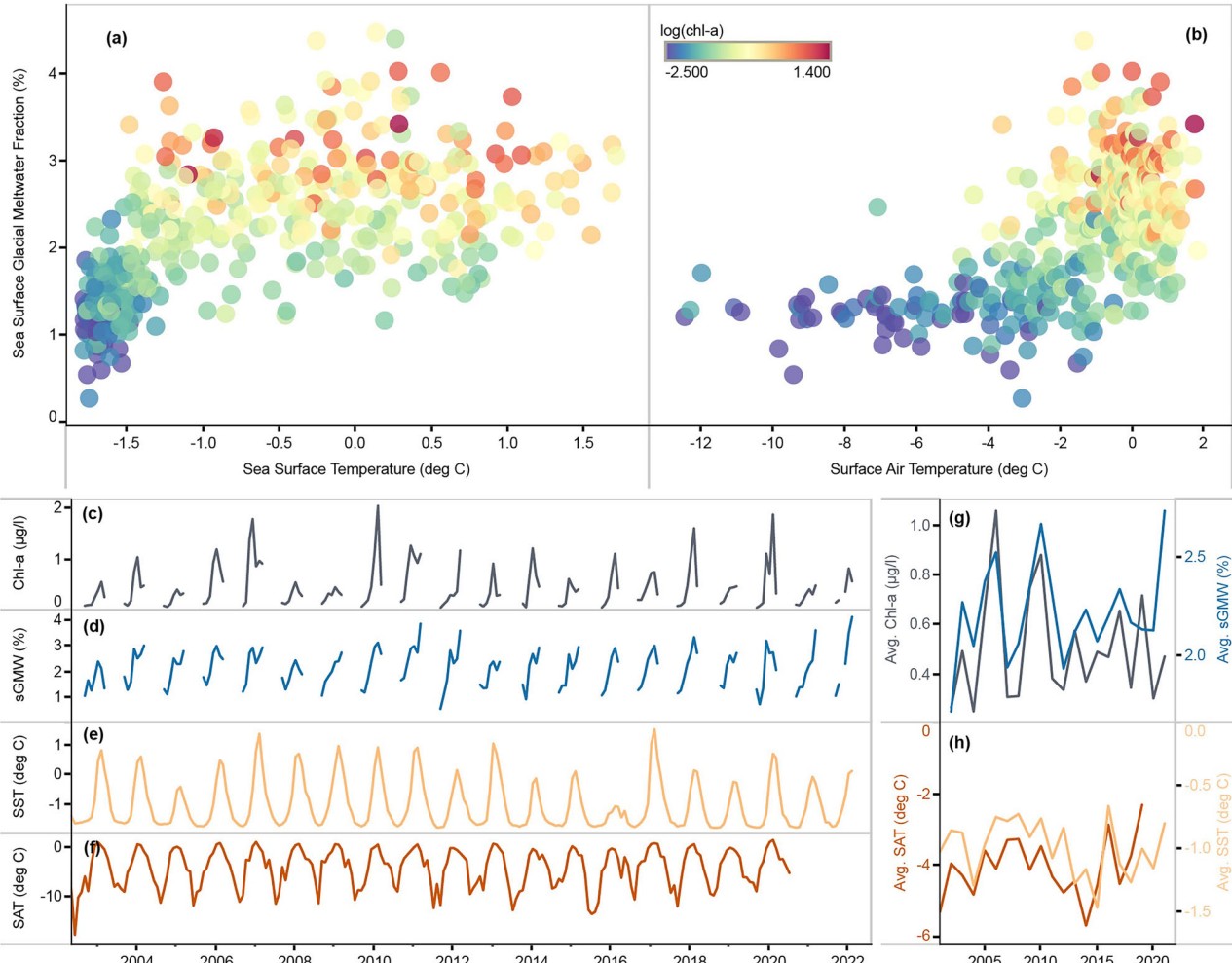

**Fig. 3 | Seasonal and interannual variabilities of environmental drivers.** Regionally averaged 8-day sGMW fraction composite compared with **a** sea surface temperature (SST), **b** surface air temperature (SAT), where the color represents chl-a concentration on a logarithmic scale. Time series illustrate interannual variabilities of monthly mean values for (**c**) surface chl-a concentration, **d** sGMW fraction, **e** SST, and **f** SAT. The gaps in chl-a concentration and sGMW fraction time series are due to missing ocean color data during winter. **g, h** Annual mean values of chl-a concentration, sGMW fraction, SST, and SAT; Ocean-color-based chl-a and sGMW values are typically from September to March each year.

This export and propagation of sGMW to the greater WAP shelf has been observed in water mass hydrography[37], surface nitrate concentration[50], and dFe concentration[36,51]. Other analyses of WAP wind data also indicate an increase in wind speed during spring in recent years, coinciding with deepened MLD[52] (Fig. 2b). These wind events, along with the hydrography of eddies, influence the residence time and transport of surface physical features, such as sGMW, along the WAP[39,53,54]. Although sGMW generally promotes near-surface stratification, wind forcing and sea-ice dynamics can dominate over the seasonally averaged MLD signal, highlighting the multifaceted controls on upper-ocean structure as well as the importance of utilizing observations across different spatial and temporal scales when studying the WAP ecosystem.

## Glacial meltwater seasonality

Chl-a concentration is correlated with sGMW discharge across multiple time scales, seasonally and interannually (Figs. 2a and 3c, d, g), while the production of sGMW is correlated with increasing sea surface temperature (SST; Fig. 3a) and increasing surface air temperature (SAT; Fig. 3b). In general, both SST and SAT increase from winter to summer. Thus, when sGMW is compared to SST and SAT, it illustrates the seasonal progression of sGMW during the phytoplankton growing season (Fig. 3a, b). Ocean temperature exerts a strong influence on subglacial and submarine melting at the glacio-marine interface, impacting the production and upwelling of

sGMW[10,15,30]. High SAT can regulate glacial melting at surface that is exposed to the atmosphere[14,55]. Together, elevated SST and SAT drive glacial meltwater production over the course of each year, with the strongest effects occurring during the summer (Fig. 3d, h). In addition to the impact of SST and SAT, circumpolar deep water is characterized by relatively high temperature (+0.1–2 °C) and salinity (34.62–34.73 PSU)[56]. Hence, the effect of warm deep ocean temperature in this region is also important to sGMW production, where meltwater released at depth near the glacial fronts can rise as buoyant plumes and thus contributes to sGMW fraction[15].

SST and SAT also exhibit strong seasonal and interannual variabilities, which drive variability in sGMW fraction. In most years, sGMW increases ahead of late autumn (Fig. 3d). This increase is likely due to glacial meltwater accumulation at the surface throughout the summer, which is consistent with in-situ measurements in Antarctic fjords, where more glacial meltwater was observed during autumn when compared with late spring[35]. A similar accumulation effect can also be observed in Marguerite Bay (lower left in Fig. 1) where the southward flow of the Antarctic Peninsula Coastal Current[57,] meets the glacial drainage of George VI Ice Shelf between Alexander Island and the WAP coastline[39,58]. The interannual variability of sGMW, in turn, drives surface chl-a concentration during spring and summer (Fig. 3c). Notably, this relationship begins to decouple prior to winter despite the accumulation of sGMW in some years. This decoupling is likely driven by declining photosynthetically active radiation (PAR) in late

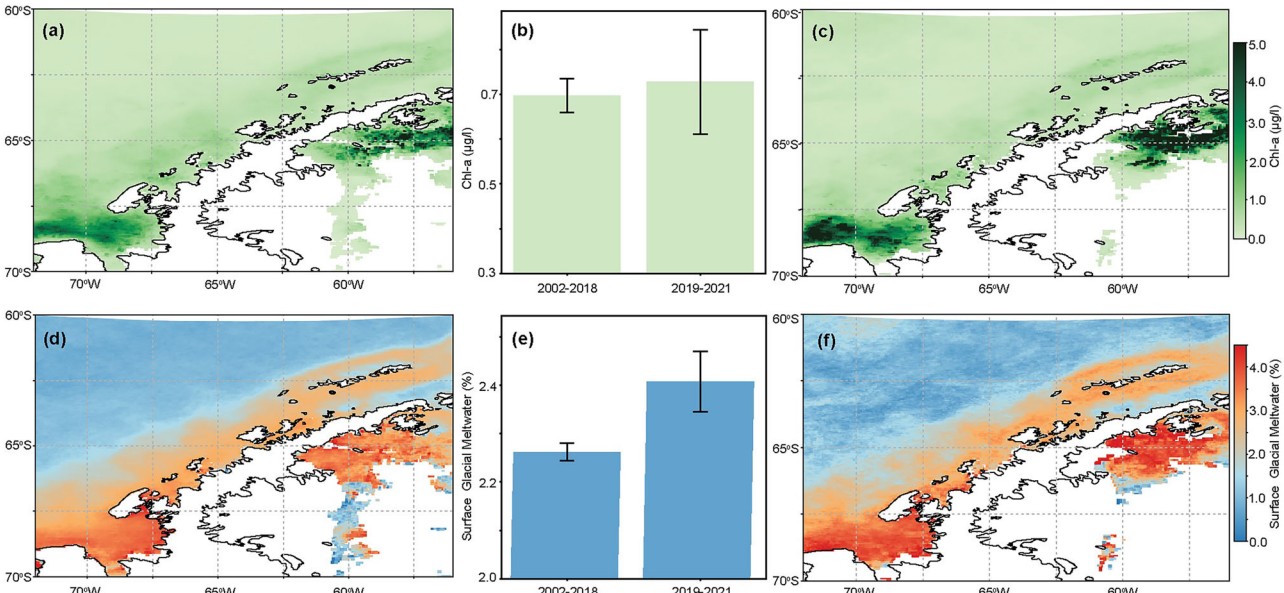

**Fig. 4 | Mean values of surface chl-a concentration and sGMW fraction between 2002–2018 and 2019–2021.** Climatological mean chl-a concentration from 2002 to 2018 (**a**) and mean values from 2019 to 2021 (**c**). The same comparison is also made for sGMW fraction (**d**, **f**). The regionally averaged values from the two time periods are also compared in the bar graph and the error bars represent standard errors of the mean (**b**, **e**).

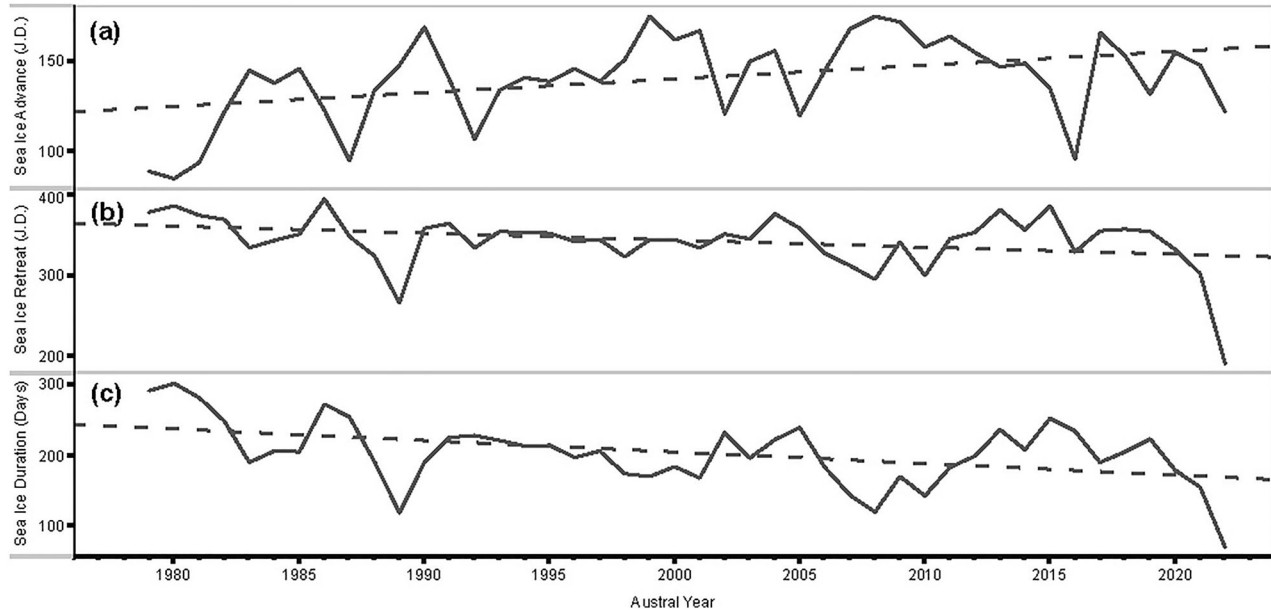

**Fig. 5 | Sea-ice annual indices over the WAP between 1979 and 2022.** Annual sea-ice advance dates (**a**) and sea-ice retreat dates (**b**) expressed in Julian days (J.D.). **c** sea-ice cover duration.

summer (shorter daylength and lower solar elevation), which reduces light availability for phytoplankton growth[28,59].

**Glacial meltwater climatology**

Comparison between the mean chl-a concentration climatologies between 2002–2018 and 2019–2021 is based on evidence of a shift in climate conditions of the WAP around 2018, characterized by changes in regional temperature, atmospheric circulation, and sea-ice patterns[60]. This division allows us to contrast the early part of our time series against the more recent period (2019–2021), potentially capturing new anomalies in glacial meltwater discharge and phytoplankton dynamics. Our results indicate that there is no significant long-term trend in chl-a concentration—albeit with

some substantial regional differences, notably in Marguerite Bay (Fig. 4). The mean WAP-regional chl-a concentration does not change significantly from 2002–2018 (0.70 ± 0.04 µg/l; Fig. 4a, b) to 2019–2021 (0.73 ± 0.12 µg/l; Fig. 4b, c). In contrast, there is a significant increase in sGMW fraction between 2002–2018 (2.52 ± 0.04%; Fig. 4d, e) and 2019–2021 (2.82 ± 0.12%; Fig. 4e, f); the observed increase in sGMW remains robust even when accounting for uncertainty estimates. Meanwhile, sea-ice along the WAP experienced a significant decline (Fig. 5). Between 1979 and 2022, the sea-ice advance date began to occur later (linear regression slope, $m = 0.76$ J.D. yr$^{-1}$, $p < 0.01$) (Fig. 5a), while sea-ice retreat date occurred earlier ($m = -0.87$ J.D. yr$^{-1}$, $p < 0.05$) (Fig. 5b), resulting in an overall shortening of the sea-ice season during this time period ($m = -1.62$ J.D.

yr$^{-1}$, $p < 0.01$) (Fig. 5c). For instance, sea-ice duration decreased drastically from 292 days in 1979 to 71 days in 2022, far exceeding the linear trend estimate (~70 days), highlighting an accelerated recent decline. Our results are consistent with the notion that glacial meltwater in the WAP can supply micronutrients, particularly iron, thereby enhancing phytoplankton growth[36,40,61]. Although this study does not include direct measurements of dFe, earlier studies demonstrate that subglacial discharge, glacier melt plumes, and melt-induced upwelling can introduce iron into the upper ocean[36,40,61–63]. Sea-ice is also recognized as a potential iron source[23], and declining sea-ice cover may diminish its fertilization role. However, our findings suggest that the net phytoplankton response remains positive in many years, possibly due to increased meltwater iron input compensates for reduced sea-ice. Further field and ongoing remote sensing efforts specifically targeting dFe data are needed to study this fertilization pathway in more detail.

Our results indicate that the impacts of sea-ice loss on surface chl-a concentration along the WAP are likely mitigated by an increase in sGMW discharge, resulting in no significant net change in overall phytoplankton biomass. However, additional mechanisms (such as changes in large-scale wind forcing or ocean temperature) may also contribute by promoting favorable mixing or earlier open-water conditions[64]. Sea-ice can seed and initiate the early-spring diatom community[19], provide a habitat for other organisms in the WAP region[1], and prevent wind-driven deep mixing[24,25]. Thus, sea-ice loss is expected to negatively impact phytoplankton communities[2]. However, the fertilization effect of additional sGMW input has likely masked the negative impact from sea-ice loss. Nevertheless, this phenomenon is expected to occur only in the short term when sGMW discharge remains at a moderate rate. Although the WAP presents unique conditions, similar meltwater-driven nutrient and stratification processes occur in other glaciated polar regions (e.g., Greenland). As the WAP continues to warm and glacial meltwater discharge intensifies over time[10,17], a negative impact on the phytoplankton community could be expected. The WAP sGMW fraction will likely increase in the future, similar to that of the Arctic and Greenland today, where heavy sediment loading associated with intense meltwater discharge impacts upper ocean light availability[65,66]. In these environments, suppressed chl-a concentration is found in the vicinity of the sGMW's source – the glacio-marine interface[67], whereas the fertilization effect is experienced by phytoplankton often farther downstream in fjords[68] or over the Greenland continental shelf[69] and open ocean[70]. Moreover, while our analysis establishes a strong correlation between sGMW fraction and surface chl-a concentration, future work should address how these relationships impact depth-integrated biomass and biogeochemical cycles. As sediment-laden meltwater becomes more common in the WAP, further study is needed to determine whether its turbidity could offset the fertilization benefits of added nutrients and ultimately impact phytoplankton productivity in these coastal and shelf waters.

The fertilization effect of increased sGMW in the WAP is likely a transient phenomenon. The "Arctification" of glacial meltwater along the WAP can already be observed at the northern end of the Peninsula, in Potter Cove situated on King George Island, where its tidewater glacier retreated and became largely a land-terminating glacier. This led to the discharge of sediment-laden glacial meltwater into Potter Cove that negatively impacted phytoplankton growth[34,71]. In the future, the acceleration of subglacial basal melting along the WAP[10,14] could profoundly alter the region's phytoplankton community composition and the broader marine ecosystem. While the input of sGMW currently appears to benefit phytoplankton by providing essential nutrients, any substantial increase in meltwater could disrupt the present ecological dynamics. Excess sGMW input can lead to significant sediment loading and reduce light availability, potentially impacting the fertilization benefit of added nutrients, and thus diminishing phytoplankton productivity. Any reduction in primary production could cascade through the food web, affecting species from zooplankton to higher trophic levels (including fish, birds, and marine mammals), with profound ramifications for the regional carbon cycle[2,4]. These potential shifts in glacial meltwater discharge necessitate a better understanding of their impact on the WAP's ecosystem and biogeochemistry.

## Methods
### Data description
Chlorophyll-a concentration (chl-a, remotely sensed near surface): chl-a concentration served as a proxy for phytoplankton biomass in this study. Remotely sensed chl-a was derived from ocean color level 3 remote sensing reflectance data (2018.0) from the Moderate Resolution Imaging Spectroradiometer on board of Aqua (MODIS-A). Data was retrieved from NASA's Ocean Biology Distributed Data Archive Center (OB.DAAC). Throughout this study, we use remotely sensed chl-a concentration as a proxy for surface phytoplankton biomass. While this approach is widely employed to capture large-scale variability, we acknowledge that satellite-derived surface chl-a neither fully represents depth-integrated biomass or primary productivity nor resolves shifts in community composition. Previous work in the WAP region[18,28] has demonstrated that surface chl-a often correlates well with integrated chl-a under diverse conditions, although this correlation can weaken in strongly stratified waters or when nutrients become limiting in the upper ocean. Our primary aim here is to leverage chl-a to assess broad, long-term trends in phytoplankton biomass.

Chlorophyll-a concentration (chl-a, field measurements, upper 5 m of the water column)[72,73]: phytoplankton biomass was also estimated based on prior field measurements (Fig. 2a). For these measurements, water samples were filtered through Whatman glass fiber filters under low vacuum, and immediately frozen at −80 °C. This procedure was followed by the extraction of the pigments using 90% acetone solution and measuring the fluorescence of each sample's supernatant with a calibrated fluorometer (10 AU Benchtop and Field Fluorometer, Turner Designs). The calculation of chl-a concentration from fluorescence was made according to Smith et al.[74] Chl-a samples were collected in Andvord Bay as part of the FjordEco campaign[35] and over the continental shelf as part of the Palmer Long Term Ecological Research (LTER) Program[19].

SST, reanalysis product, near surface[75]: the NOAA 0.25° daily optimum interpolation sea surface temperature (OISST) was constructed by combining bias-adjusted observations from different platforms (satellite, ships, buoys) on a regular global grid, with gaps filled in by interpolation. Satellite data from the Advanced Very High Resolution Radiometer (AVHRR) provides the main input which allows high spatial-temporal coverage from late 1981 to the present.

SAT, reanalysis product, near surface[76]: ERA5 is the fifth-generation European Centre for Medium-Range Weather Forecasts (ECMWF) atmospheric reanalysis of the global climate. Reanalysis combines model data with observations from across the world into a globally complete and consistent dataset at 0.25°. ERA5 air temperature at 2 m daily data, denoted as SAT in this study, provide aggregated values for each day and the daily aggregates are calculated based on the ERA5 hourly values.

sGMW fraction, remotely sensed near surface[39]: an ocean color data product developed by Pan et al.[39] for remotely quantifying sGMW fraction in the WAP region. This ocean-color-based model was trained and evaluated against one of the most comprehensive in-situ stable oxygen isotope (δ$^{18}$O) dataset complied from the WAP region[48,58,77,78]. The model derives sGMW fraction from MODIS-A level 3 remote sensing reflectance data. The mean sGMW fraction increase from 2002–2018 to 2019–2021 presented some uncertainties (Fig. 4). These likely reflect the propagation of errors in remote sensing reflectance and meltwater end-member calibration. However, the observed increase in sGMW remains robust even when accounting for upper-bound uncertainty estimates. See "Methods" presented by Pan et al.[39] for more details.

In-situ glacial meltwater fraction (upper 5 m of the water column)[48,58,77,78]: in-situ glacial meltwater fraction is inferred from δ$^{18}$O field measurements in units of ‰. sGMW fraction is derived based on mass balance calculations according to established protocols from prior studies, and it is measured in units of freshwater %. Each discrete δ$^{18}$O sample is paired with its corresponding surface salinity value. The mass balance

calculation presumes each sample is composed of a simple mixture of three components—ocean water (ow), sea-ice meltwater (sim), and meteoric water (met), with the latter term being the sum of precipitation and glacial meltwater:

$$F_{sim} + F_{met} + F_{ow} = 1$$

$$S_{sim} \cdot F_{sim} + S_{met} \cdot F_{met} + S_{ow} \cdot F_{ow} = S_{total}$$

$$\delta^{18}O_{sim} \cdot F_{sim} + \delta^{18}O_{met} \cdot F_{met} + \delta^{18}O_{ow} \cdot F_{ow} = \delta^{18}O_{total}$$

This system of equations is solved for $F_{sim}$, $F_{met}$, and $F_{ow}$, which are the respective fractions of the three components in each sample. $S_{sim}$, $S_{met}$, and $S_{ow}$ are the salinity values for the end-member source components, while $\delta^{18}O_{sim}$, $\delta^{18}O_{met}$, $\delta^{18}O_{ow}$ are the corresponding $\delta^{18}O$ values. The sampling locations of this dataset are the Palmer LTER grid, Palmer Station on Anvers Island, Potter Cove, and Rothera Point on Adelaide Island. See "Methods" from these prior studies for more detail[48,58,77,78].

MLD, field measurement[24]: MLD calculation followed an approach described by Carvalho et al.[79] For each profile through the water column, surface MLD is estimated by finding the depth of the maximum water column buoyancy frequency ($N^2$). In addition, a quality index (QI) filter is also applied to identify water columns with no clear MLD. QI equation used in this dataset was developed by Lorbacher et al.[80] to evaluate individual MLD calculations against water column density and filter out profiles where MLD could not be resolved. The QI index evaluates the quality of the MLD calculation. Using this method, MLDs can be characterized into estimates determined with certainty ($QI > 0.8$), determined with some uncertainty ($0.5 < QI < 0.8$) or not determined ($QI < 0.5$). For the MLD dataset in this study, a QI of 0.5 was used to warrant a MLD calculation. This determination of MLD is based on the principle that there is a near-surface layer characterized by quasi-homogeneous properties and where the standard deviation of the property within this layer is close to 0. This method does not consider the strength of stratification, but only the surface layer's homogeneity. Therefore, the MLD estimate is close to the lower boundary of that vertically uniform layer. This method has been validated for locations across the Southern Ocean (including the WAP) and its ecological relevance was confirmed against discrete chl-a measurements[24,79].

Sea-ice annual indices[81,82]: satellite measurements of sea-ice concentration are from NASA's Scanning Multichannel Microwave Radiometer (SMMR) and the Defense Meteorological Satellite Program's Special Sensor Microwave/Imager (SSM/I). The indices calculation uses the GSFC Bootstrap[83] SMMR-SSM/I (quasi) daily time series that minimizes the differences between the various SMMR and SSM/I sensors[84]. The EOS Distributed Active Archive Center (DAAC) at the National Snow and Ice Data Center (University of Colorado at Boulder, https://nsidc.org) provided the every-other-day SMMR and the daily SSM/I time series.

Annual sea-ice advance date, retreat date, and ice season duration are extracted from (quasi) daily data of SMMR-SSM/I sea-ice concentration. Sea-ice advance and retreat dates are identified from an annual search window, defined such that it begins and ends during the mean summer sea-ice extent minimum in mid-February (i.e., begins Julian day 46, ends Julian day 410, or 411 in a leap year). Within this search window, advance date is identified when sea-ice concentration first exceeds 15% (i.e., the approximate ice edge) for at least 5 days. Retreat date is identified when sea-ice concentration remains below 15% until the end of the search period. If sea-ice never departed from a particular region, then day of advance and retreat are set to the lower and upper limits—Julian day 46 and 410 or 411, respectively. Ice season duration is simply the time elapsed between advance date and retreat date each year[82].

Map (Fig. 1): the climatological positions of the major fronts are estimated from observed temperature and salinity data based on Orsi et al.[85] sGMW overlay from September 2021 to February 2022 is based on aforementioned method for remotely deriving sGMW fraction[39].

## Data processing and analysis

Remotely sensed chl-a concentration, SST, SAT, and sGMW fraction data were accessed via Google Earth Engine using a Python package, geemap, for interactive mapping[86]. Figure 1 was generated using Quantarctica, an integrated mapping environment for the Southern Ocean[87].

The region of interest (ROI) in this study coincides with the Palmer LTER grid shown in Fig. 1. The nodes of the ROI are at longitude/latitude of −66.86°/−63.97°, −78.48°/−68.09°, −76.14°/−69.24°, −66.86°/−63.97°, with the coast-facing edge of the ROI extending to the shoreline. Regionally averaged values presented in this study are based on this ROI.

Annual averages in this study are based on "austral years," which are defined as September 1st to the following year August 31st to capture the full seasonal cycle of phytoplankton growth—from austral spring to winter. This definition avoids averaging by calendar years, which separates the austral growing season. The austral year definition is only relevant in Fig. 2b, where each data point represents an austral year summer average, in Fig. 3g, h the annual averaged values, and also in the labels in Fig. 4, albeit the data were only from the summer months at the peak of most year's growing season. Summer anomalies, such as those presented in Fig. 2, are calculated based on: $\frac{(\text{summer mean} - \text{climatological mean})}{\text{climatological mean}} \times 100\%$, where the summer mean is calculated based on austral years, averaging values from December, January, and February.

The temporal grouping presented in section "Glacial Meltwater Climatology" and in Fig. 4 are based on Turner et al.[60] who determined a cooling period from 1998 to 2018 in the Antarctic Peninsula given by the stacked and normalized SAT records from various Antarctic stations, hence a transition in the region's climate around this time[60,88]. The regionally averaged summer (D/J/F) daily data is separated into two groups—2002–2018 and 2019–2021. A statistical test was used to determine if the two time periods were significantly different. Assumptions regarding the two groups were first checked to determine what type of statistical test was appropriate. First, the two time periods have independent observations. Secondly, the Shapiro–Wilk test was used to determine normality of each group[89]. Thirdly, homoscedasticity between the two groups was determined using the Levene's test[90]. For remotely sensed chl-a concentration, the Shapiro–Wilk test on the 2002–2018 group has a p value of 0.00, and the 2019–2021 group has a p value of $1.34 \times 10^{-21}$, indicating both groups are not normally distributed. The Levene's test on the two groups results in a p value of 0.99, indicating variances are equal. For the remotely sensed sGMW fraction, the Shapiro–Wilk test on the 2002–2018 group has a p value of $8.36 \times 10^{-6}$, and the 2019–2021 group has a p value of 0.21, indicating the former group is not normally distributed. The Levene's test on the two groups result in a p value of 0.03 indicating variances are not equal. The issue with groups' normality is mitigated given the sample size, where the 2002–2018 period has 1235 data points and the 2019–2021 period has 192. Furthermore, the Welch's t-test was chosen to test for difference between the means of the two groups, which account for unequal variances and unequal sample sizes[91]. The Welch's t-test on the chl-a datasets result in a Welch's t-statistic of −0.53 and p value of 0.60, indicating no significant difference in chl-a concentrations between the two time periods. The Welch's t-test on the sGMW datasets result in a Welch's t-statistic of −3.89 and p value of 0.0001, indicating a significant difference in sGMW fractions between the two time periods.

## Reporting summary

Further information on research design is available in the Nature Portfolio Reporting Summary linked to this article.

## Data availability

All databases and datasets used in this study are cited in the Methods section. Satellite-based remote sensing ocean color data can be retrieved from NASA's OB.DAAC at the Goddard Space Flight Center (https://oceancolor.gsfc.nasa.gov/). In-situ data from Andvord Bay can be found at the U.S. Antarctic Program Data Center (USAP-DC, https://doi.org/10.15784/

601158). Datasets from the Palmer LTER Program are archived at https://pallter.marine.rutgers.edu/data/. Optimum Interpolation SST (OISST) can be retrieved from the National Oceanic and Atmospheric Administration's National Centers for Environmental Information (NOAA NCEI, https://www.ncei.noaa.gov/products/optimum-interpolation-sst). ERA5 hourly data is based on the European Centre for Medium-Range Weather Forecasts (ECMWF) reanalysis product: https://cds.climate.copernicus.eu/datasets/reanalysis-era5-single-levels?tab=download. Geographic and oceanographic information presented in Fig. 1 is based on data retrieved through Quantarctica (https://npolar.no/en/quantarctica/).

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

## Acknowledgements

The corresponding author carried out this research at the Jet Propulsion Laboratory, California Institute of Technology, under a contract with the National Aeronautics and Space Administration (NASA). S.S. and O.S. were supported by the National Science Foundation, Office of Polar Programs (OPP-1440435 and OPP-2026045) and the NASA Interdisciplinary Science (IDS) Program, Award Number 80NSSC24K0481. The participation of M.P.M. was funded by the Natural Environment Research Council via award NE/W004933/1 (BIOPOLE). R.A.R. and M.V. were supported by the NASA Citizen Science Earth System Program (CSESP), Award Number 80NSSC22K1914. F.A.H. was supported by the European Union (ERC, VERTEXSO, 101041743) and the Initiative and Networking Fund of the Helmholtz Association (Grant Number: VH-NG-19-33). A.O. was supported by the NASA Biodiversity and Ecological Conservation Program, Award Number 80NSSC25K7240. The corresponding author and C.E.M. were also supported by the NASA Interdisciplinary Science (IDS) Program. Any opinions, findings and conclusions or recommendations expressed in this material are those of the authors and do not necessarily reflect the views of NASA.

## Author contributions

B.P.: conceptualization, data curation, formal analysis, investigation, methodology, project administration, software, validation, visualization, writing—original draft, writing—review and editing; M.M.G.: conceptualization, formal analysis, funding acquisition, methodology, project administration, resources, supervision, validation, writing—original draft, writing—review and editing; S.S.: data curation, formal analysis, investigation, methodology, resources, validation, writing—original draft, writing—review and editing; O.S.: data curation, formal analysis, investigation, methodology, resources, validation, writing—review and editing; M.P.M: data curation, formal analysis, investigation, methodology, resources, validation, writing— review and editing; R.A.R.: formal analysis, investigation, methodology, validation, writing—original draft, writing—review and editing; M.V.: investigation, resources, validation, writing—review and editing; F.A.H.: methodology, validation, writing—review and editing; A.J.O.: data curation, software, writing—review and editing; C.E.M.: funding acquisition, supervision, writing—review and editing.

## Competing interests

The authors declare no competing interests.

## Ethics approval and consent to participate

This study was conducted in accordance with the principles of inclusivity and ethical research practices.
