## [Peer Review file · Communications Earth & Environment]

Impact of Glacial Meltwater on Phytoplankton Biomass along the Western Antarctic Peninsula

Corresponding Author: Dr B. Jack Pan

Version 0:

Decision Letter:

Dear Dr Pan,

Please accept our sincere apologies for the delay in sending a decision on your manuscript. Your manuscript titled "Impact of Glacial Meltwater on Phytoplankton Ecology along the Western Antarctic Peninsula" has now been seen by 3 reviewers, and we include their comments at the end of this message. They find your work of interest, but some important points are raised. We are interested in the possibility of publishing your study in Communications Earth & Environment, but would like to consider your responses to these concerns and assess a revised manuscript before we make a final decision on publication. Specifically, a revised manuscript must:

1. Provide a more detailed explanation of the glaciological and ecological mechanisms driving the observed patterns, particularly in relation to glacier types, sediment transport, and sea ice algae.
2. Provide compelling support for the claims regarding glacial meltwater stabilizing the surface ocean by revisiting the discussion of mixed layer depth and ensuring that the arguments are consistent and well-supported by data

We therefore invite you to revise and resubmit your manuscript, along with a point-by-point response that takes into account the points raised. Please highlight all changes in the manuscript text file.

Please submit your point-by-point responses as a separate file, distinct from your cover letter where you can add responses to the Editors' comments that you do not want to be made available to the reviewers. Word files are preferred. We recommend that any figures, tables or graphs that are included in the response to reviewers are also included in the main article or Supplementary Information.

Please use the following link to submit your revised manuscript, point-by-point response to the referees' comments (which should be in a separate document to any cover letter), a tracked-changes version of the manuscript (as a PDF file) and the completed checklist:

Link Redacted

We hope to receive your revised paper within six weeks; please let us know if you aren't able to submit it within this time so that we can discuss how best to proceed. If we don't hear from you, and the revision process takes significantly longer, we may close your file. In this event, we will still be happy to reconsider your paper at a later date, as long as nothing similar has been accepted for publication at Communications Earth & Environment or published elsewhere in the meantime.

Please do not hesitate to contact us if you have any questions or would like to discuss these revisions further. We look forward to seeing the revised manuscript and thank you for the opportunity to review your work.

Best regards,

Dr. Dania Albin PhD
Editorial Board Member
Communications Earth & Environment
orcid.org/0000-0003-4236-1536

Alice Drinkwater, PhD
Associate Editor
Communications Earth & Environment

EDITORIAL POLICIES AND FORMATTING

Editorial Policy: [Policy requirements](https://www.nature.com/documents/nr-editorial-policy-checklist.pdf) (Download the link to your computer as a PDF.)

- Behavioural and social science
- Ecological, evolutionary & environmental sciences
- Life sciences

<https://www.nature.com/documents/nr-reporting-summary.zip>

Furthermore, please align your manuscript with our format requirements, which are summarized on the following checklist: [Communications Earth & Environment formatting checklist](https://www.nature.com/documents/commsj-phys-style-formatting-checklist-article.pdf)

and also in our style and formatting guide [Communications Earth & Environment formatting guide](https://www.nature.com/documents/commsj-phys-style-formatting-guide-accept.pdf).

***** DATA:** Communications Earth & Environment endorses the principles of the Enabling FAIR data project (<http://www.copdess.org/enabling-fair-data-project/>). We ask authors to make the data that support their conclusions available in permanent, publically accessible data repositories. (Please contact the editor if you are unable to make your data available).

All Communications Earth & Environment manuscripts must include a section titled "Data Availability" at the end of the Methods section or main text (if no Methods). More information on this policy, is available at <http://www.nature.com/authors/policies/data/data-availability-statements-data-citations.pdf>.

If a community resource is unavailable, data can be submitted to generalist repositories such as [figshare](https://figshare.com/) or [Dryad Digital Repository](http://datadryad.org/). Please provide a unique identifier for the data (for example a DOI or a permanent URL) in the data availability statement, if possible. If the repository does not provide identifiers, we encourage authors to supply the search terms that will return the data. For data that have been obtained from publically available sources, please provide a URL and the specific data product name in the data availability statement. Data with a DOI should be further cited in the methods reference section.

REVIEWER COMMENTS:

Reviewer #1 (Remarks to the Author):

Here, Pan et al. present analysis linking sea ice coverage, glacial melt water, and phytoplankton biomass, finding that, as sea ice coverage in the West Antarctic Peninsula has declined since 1979, phytoplankton biomass has stayed high due to glacial melt water providing nutrients and stratifying the surface ocean. This manuscript is well written throughout and the data are well presented, and I find the conclusions of a link between phytoplankton biomass and glacial melt water convincing. However, there are two major points in the paper, both relating to MLD, that require clarification. First, the authors state that one of the primary reasons behind the correlation between phytoplankton biomass and glacial melt water concentration is that the glacial melt water helps to stratify the water column, providing more light to phytoplankton. However, their results indicate that years with high glacial melt water can have shallower or deeper MLDs. The authors argue that MLD is a transient/snapshot in time metric in the body of the paper, but claim in the abstract and conclusions that GMW is associated with more stable/shallower MLDs, a claim which isn't supported by their results. Looking at their results (Fig. 2b specifically), it seems to me that sea ice duration is the greatest control on MLD - years with anomalously long SI duration tend to have shallower MLDs, while years with short SI duration have deeper MLDs. Secondly, since this paper presents surface Chl concentration and not integrated primary production, slightly more care should be taken in claims of the impacts of glacial melt water on phytoplankton communities and the ecosystem. If MLD is shallower in one year, a large change in phytoplankton biomass over a thinner layer of the ocean may not correspond to any change in phytoplankton productivity (or could even coincide with a reduction in NPP).

Minor comments:

Line 25-26: These productivity rates (from Vernet et al., 2008) are averaged over the summer/observation period, not the year. I believe the second paper cited just references the first paper cited and does not provide measured NPP.

Fig 2b: The Chl-a anomaly legend is difficult to understand. Consider showing the full size of the dots (not just horizontally but also vertically) and eliminating the darker grey circle (I think this is intended to show how much larger each circle is from the previous one, but I find it confusing). I'm also confused with this plot/caption - how is it that only one of the 17 years represented here has a glacial melt water concentration anomaly below the 2002-2022 climatology?

Lines 70-72: High Chl concentration and long sea ice duration can co-occur, but they certainly do not look correlated in Fig 2b - and thus I don't think these data convince me that 'longer sea ice duration...can lead to high Chl'. While sea ice presence can prevent wind mixing and ice melt can provide nutrients and stabilize the ocean surface, a shorter sea ice duration can also alleviate light limitation, making the relationship between sea ice and productivity more complicated than described here (but more in line with the data presented, whereby shorter sea ice duration can coincide with higher Chl).

Lines 86-88: I think here the authors again note a co-occurrence and subsequently (in the abstract and concluding paragraph) conflate that co-occurrence with causality. Looking at Fig 2b, I would argue that it appears that sea ice duration is the greatest control on MLD, not GMW, but that GMW is a greater control on Chl concentration than sea ice concentration.

Line 90: I would replace the end of this sentence with "offshore onto the wider continental shelf."

Line 99: I don't like the use of 'coupled' here and would replace with correlated, for example.

Line 129: A sentence discussing why these time periods (2002-2018 and 2019-2021) were chosen for analysis would be helpful.

Line 137: I would love to see these trends quantified in d yr^{-1} , especially since it's hard to compare them in Figure 5 due to the different scales of the y axes.

Lines 144-147: I believe the first part of the first sentence here is referring to sea ice algae. While there can be very high concentrations of SIA in the Southern Ocean, they are pretty unimportant in terms of total NPP. They are critical in terms of timing of the availability of primary production, but that's not really discussed elsewhere in this paper. This sentence and the next appear to conflate SIA and phytoplankton, so I might rephrase/omit to avoid confusion.

Line 164: Again, I'm not certain that your analysis demonstrates that glacial melt water stabilizes the surface ocean.

Reviewer #2 (Remarks to the Author):

The study of Pan et al. examines the ecological impact of glacial meltwater on phytoplankton along the Western Antarctic Peninsula, using 20 years of multi-source data. The results reveal a correlation between meltwater and phytoplankton biomass, that is suggested to be driven by nutrient fertilization and surface ocean stabilization. The findings suggest that increased glacial meltwater may mitigate the negative effects of sea-ice loss on phytoplankton productivity, emphasizing the importance of meltwater as a critical driver in polar coastal ecosystems.

The results, based on Pan et al. previous work using remote sensing for estimating surface glacial melt water, are quite convincing and certainly relevant for a broad research community studying the impacts of global change in polar regions. However, the complexity of ice-ocean-ecosystem interactions requires more thorough conceptual exposition. While the discussion touches on these aspects, they are not adequately framed earlier in the manuscript, leaving the discussion somewhat fragmented.

My suggestion is to (in the introduction) describe the system better than it is currently with specific attention to:

The sea ice: Elaboration on the process "seeding" (Line 36-38) and the foundational role of sea-ice algae in initiating blooms by providing the initial population of diatoms and contributing to the conditions needed for their proliferation – including their role as Fe and other nutrient 'keepers'.

The glaciological setting of glaciers in the WAP and the mechanisms involved in their regulation of oceanic nutrients: This is

critical to understanding their impact on primary productivity. A key question is which glacier types—marine-terminating or land-terminating—are most relevant in this region. The mechanisms through which these glacier types deliver meltwater to the ocean differ significantly and have direct implications for nutrient content. For marine terminating WAP glaciers, what is the partitioning between surface melting, calving, and submarine melting? This needs to be introduced and explained to provide context, as it is important for linking meltwater production to ocean fertilization processes. Specifically, the surface melt, and hence subglacial discharge, is important to understand for WAP, as it can influence phytoplankton growth by:

1. Upwelling of nitrate-rich waters: Subglacial discharge promotes upwelling, bringing deep, nutrient-laden waters into the photic zone, where phytoplankton can use it.

2. Sediment transport: Subglacial processes determine the extent and composition of sediments transported to the ocean, impacting nearshore and offshore ecosystems.

These mechanisms should be briefly explained. I am sure the authors are well aware of all of this, but it does not come out clearly in the current manuscript.

Another question in this regard is also relevant to address: To what extent do WAP glaciers produce sediment? It is likely limited, given historically low surface melt in the region, which might not have primed glacier beds for substantial soft sediment production. Furthermore, the bioavailability of these sediments for primary producers remains an open question. While the discussion suggests sediment increases in the future could limit sunlight in coastal systems, it is unclear how this might impact offshore ecosystems.

Although the authors cannot resolve all these unknowns, they should describe these system complexities better in the introduction. This will allow their findings—i.e. the observed increase in phytoplankton growth despite low sea-ice years alongside higher glacial meltwater input—to fit more effectively into the broader puzzle of WAP ecosystem dynamics. By framing their study within a better outline of the understanding of the WAP glaciomarine environment, they can enhance the impact of their specific data and conclusions and provide recommendations for future studies in the region.

Some additional comments:

Fig. 2 Add details such as the time period and data type directly to Figure 2, along with clear color references. For MODIS data, include references to methodologies used to obtain these estimates.

Line 44-49: The previous work of the main author in the region is outlined (Pan et al. Environmental drivers of phytoplankton taxonomic composition in an 357 Antarctic fjord. *Prog. Oceanogr.* 183, 102295 (2020) and Pan, et al. Remote sensing of sea surface glacial meltwater on the Antarctic Peninsula shelf. 1–16 (2023) doi:10.3389/fmars.2023.1209159.). I assume that the novelty of the study here refers to the longer time period under study. However, I cannot confirm this without reviewing what time periods the two earlier papers by the author covers. The novelty could easily be clarified with more precise wording in line 44-47, and line 49.

Line 59: Revise the section header to better reflect its content, e.g., 'Glacial meltwater drives offshore phytoplankton growth'.

Line 81: The authors could start this section by very briefly explaining (1-2 sentences) the relevance of the MLD for the primary producers (i.e. regulating nutrient and light availability etc). It will provide better context for the further discussion in the same section (line 81-96).

Line 131: To add clarity provide reference to Fig 4 for this statement.

Line 134-136: The uncertainties on the sGMW fractions are quite low, and there is no mentioning in the manuscript how the uncertainties are assessed. If they were higher, any robustness of the observed increase (2.52% to 2.82%) would be less convincing.

In the method section: Spell out abbreviations and provide error estimates for the key metrics.

Reviewer #3 (Remarks to the Author):

General comments

The manuscript by Pan et al. examines the influence of sea surface glacial meltwater (sGMW) on chlorophyll-a (Chl-a) dynamics along the Western Antarctic Peninsula (WAP). Utilizing a dataset spanning two decades, the study suggests that glacial meltwater influences phytoplankton biomass through nutrient enrichment, surface stabilization, and light optimization. The findings propose that sGMW mitigates the adverse effects of sea ice loss on primary production, offering insights into polar ecosystem resilience under climate change.

The study addresses a timely aspect of climate change in polar regions, providing insights into the role of glacial meltwater as a biogeochemical driver in a rapidly changing ecosystem. It effectively combines remote sensing, reanalysis products, and in-situ observations and is well-written.

However, several concerns need to be addressed:

-Scope of Phytoplankton Ecology: The manuscript promises to focus on impacts on phytoplankton ecology; however, the only biogeochemical variable explored is chlorophyll-a. There are references to phytoplankton biomass, but no explicit biomass data are reported. Changes in Chl-a concentration can also result from shifts in community composition, which are not discussed.

-Nutrient Fertilization Attribution: The attribution of changes to nutrient fertilization, particularly iron, is plausible but not substantiated with direct evidence. Furthermore, if sea ice serves as an iron source, its decline would also reduce its fertilization effect.

-Novelty of Findings: Lines 43-44 indicate that sGMW has already been identified as a driver for phytoplankton productivity in the WAP region, potentially limiting the novelty of this work.

-Geographical Focus: The manuscript exclusively focuses on the WAP. A broader discussion on whether these findings can be extrapolated to other polar regions would enhance the study's relevance.

Overall, while the topic is interesting and timely, additional data and analyses are necessary to strengthen the work. I wonder if these would be available from the Palmer LTER transects?

Specific Comments

Title: The term "phytoplankton ecology" is not adequately addressed in the manuscript. Consider revising the title to reflect the study's actual focus.

Abstract:

Line 13: Replace "phytoplankton biomass" with "Chl-a" to accurately represent the measured variable.

Main Text:

Line 24: Consider removing the word "polar" for conciseness.

Lines 38-39: Expand on the concept of ice algae seeding to provide clarity.

Line 41: Rephrase "Sea ice ...hosting under-ice diatoms..." for clarity, as sea ice hosting under-ice species is contradictory.

Lines 46-47: Elaborate on these findings, particularly regarding community composition.

Lines 54-56: Revise this section, as the findings do not clearly support the stated "ecological impact."

Lines 68-69: Clarify how these observations enhance understanding of sGMW's impact on marine ecology.

Lines 70-73: Expand on the references provided, as the statement in lines 70-72 is not universally applicable.

Lines 77-78: Acknowledge that Chl-a concentrations are not always indicative of phytoplankton abundance; please revise.

Line 79: Replace the word "ecological" with a more appropriate term.

Lines 93-94: An analysis of wind patterns would enhance the robustness of the study; consider incorporating this aspect.

Line 107: This statement is self-evident; consider rephrasing or removing.

Line 125: Clarify what is meant by "unfavourable light conditions."

Line 132: Using the mean may not be informative; consider analyzing seasonal maxima in relation to ice season duration and sGMW for more insightful information.

Line 134: If the result is significant, report the p-value.

Lines 142-144: Provide further evidence or explore alternative explanations for the stated results.

Lines 151-153: This point is intriguing and warrants further development.

Lines 166-170: Expand this section to provide more depth.

Line 181: Phytoplankton abundance is not depicted in Fig. 2a; please revise accordingly.

Figures:

Figure 1: Including additional information on currents and winds would enhance the figure's informative value.

Figure 2: There is no data related to "phytoplankton abundance" presented; please revise the figure or its description.

Grammar and Clarity Corrections

Replace terms like "ecological" with more precise language relevant to the context.

Clarify statements that are currently vague or self-evident to enhance readability and comprehension.

Communications Earth & Environment is committed to improving transparency in authorship. As part of our efforts in this direction, we are now requesting that all authors identified as 'corresponding author' create and link their Open Researcher

and Contributor Identifier (ORCID) with their account on the Manuscript Tracking System prior to acceptance. ORCID helps the scientific community achieve unambiguous attribution of all scholarly contributions. You can create and link your ORCID from the home page of the Manuscript Tracking System by clicking on 'Modify my Springer Nature account' and following the instructions in the link below. Please also inform all co-authors that they can add their ORCIDs to their accounts and that they must do so prior to acceptance.

Version 1:

Decision Letter:

Dear Dr Pan,

Your manuscript titled "Impact of Glacial Meltwater on Phytoplankton Growth along the Western Antarctic Peninsula" has now been seen by our reviewers, whose comments appear below. In light of their advice we are delighted to say that we are happy, in principle, to publish a suitably revised version in Communications Earth & Environment.

We therefore invite you to revise your paper one last time to address the remaining concerns of our reviewers. At the same time we ask that you edit your manuscript to comply with our format requirements and to maximise the accessibility and therefore the impact of your work.

In particular, please ensure that all of Reviewer 3's concerns are fully addressed. While they commend the clarifications around the use of chlorophyll-a and the expanded discussion, they raise some concerns. These include the need to temper causal claims related to nutrient fertilization from meltwater, acknowledging the absence of direct nutrient measurements, and considering alternative mechanisms such as upwelling or benthic fluxes. Reviewer 3 also recommends clearer differentiation between correlation and causation, more precise use of statistical language, clarification of chlorophyll-a as a proxy for biomass rather than abundance, a brief forward-looking note on phytoplankton community composition, and correction of minor language issues throughout the manuscript.

EDITORIAL REQUESTS:

****Please take care to match our formatting and policy requirements. We will check revised manuscript and return manuscripts that do not comply. Such requests will lead to delays. ****

SUBMISSION INFORMATION:

OPEN ACCESS:

Communications Earth & Environment is a fully open access journal. Articles are made freely accessible on publication. For further information about article processing charges, open access funding, and advice and support from Nature Research, please visit <https://www.nature.com/commsenv/open-access>

Link Redacted

**** This url links to your confidential home page and associated information about manuscripts you may have submitted or be**

reviewing for us. If you wish to forward this email to co-authors, please delete the link to your homepage first **

Best regards,

Alice Drinkwater, PhD
Associate Editor
Communications Earth & Environment
Consulting Editor
Communications Sustainability

REVIEWERS' COMMENTS:

Reviewer #1 (Remarks to the Author):

The authors have satisfied my major concerns with their edits to this work, and I appreciate their efforts to revise the manuscript. However, I don't understand why reporting the slopes (in days per year) of the linear regressions they present in Fig. 5 is "beyond the scope of this work." Including only the number of sea ice covered days in 1979 and 2022 gives readers a very skewed sense of the linear regression of the timeseries, which shows a far more moderate trend. I would also revise L124 - if the aim is to talk about the potential that sea ice algae seed phytoplankton blooms, phrasing such as "facilitates the accumulation of sea ice algae that may seed phytoplankton blooms" would be clearer.

Reviewer #2 (Remarks to the Author):

The authors have done a very thorough job in complying with the reviewers queries, concerns and suggestions, and the manuscripts i markedly improved. I have no further comments.

Reviewer #3 (Remarks to the Author):

Second-Round Review of "Impact of Glacial Meltwater on Phytoplankton Ecology along the Western Antarctic Peninsula"

Please note that line numbers refer to the marked-up revised ms.

General comments

Overall, the revised manuscript has improved and addresses some key concerns raised in the first-round review. The authors have clarified their use of chlorophyll-a as a proxy for biomass (but not for abundance, see my comment below), added supporting evidence for the proposed nutrient fertilization mechanism, and expanded the discussion to place their findings in a broader polar context. The data analysis and the conclusions, while based largely on correlations, are more convincing given the additional literature support. However, I still think that providing additional data directly in the ms would make it stronger (e.g., winds, see my recommendations further) as well as still working on revising causality for nutrient fertilization.

While the correlation between Chl-a and sGMW is compelling, the inference that meltwater acts primarily through nutrient fertilization, especially via iron, remains speculative. No direct nutrient data (e.g., Fe, NO₃⁻, PO₄³⁻, Si(OH)₄) are presented. The manuscript discusses literature findings from other regions (e.g., fjords, Andvord Bay), but these cannot be assumed to be representative of the broader WAP shelf over the 20-year period studied here. Similarly, potential contributions from upwelling, lateral advection, or benthic fluxes are not considered. While some of these are mentioned in the discussion, they are not integrated into the data analysis or explicitly tested. The revised manuscript improves clarity around this issue and now refers to nutrient input as a "likely" mechanism in some places. However, other sections still present this conclusion too strongly. For instance, the manuscript states that the strong positive correlation between sGMW and Chl-a points to nutrient enrichment from glacial meltwater. This causal language should be revised to make clear that the mechanism is hypothetical and not directly measured in this study.

One suggestion is for the authors to ensure clarity wherever they discuss correlations: differentiating clearly between what their data show (correlation between melt and Chl-a) and what they hypothesize as the cause (iron/stability), backed by prior studies. A sentence or two explicitly framing it as a "likely mechanism supported by [literature]" would guard against readers over-interpreting correlation as causation.

Whenever authors talk about anything "significant" (or "not significant"), they need to specify to what degree (e.g. with p-values). Please review this throughout the text or change your wording accordingly.

While Chlorophyll-a concentration can more acceptably be used (upon limitations acknowledgement) as a proxy for phytoplankton biomass, caution is warranted when interpreting Chl-a as a direct proxy for cell abundance. In general, Chl-a

is a poor indicator of abundance due to physiological, taxonomic, and environmental variability that can bring the same Chl-a concentrations to represent completely different abundances. This is widely acknowledged in the literature. I invite the authors to reflect on this and modify the entire text accordingly, including the title, by pointing to Chl-a as a proxy for biomass rather than abundance.

In the Discussion or Conclusions, it might be valuable to include one sentence about phytoplankton community composition as a future consideration. While the study focuses on total biomass, the authors have the expertise to note that the ecological impact of meltwater could also involve shifts toward certain taxa (for example, meltwater has been associated with smaller cryptophytes in some WAP fjords, whereas diatoms dominate in stratified but nutrient-rich conditions). A brief mention that the community structure and broader food-web implications were not directly assessed here and remain important topics for future research would acknowledge the “ecology” aspect. Such a statement would add a forward-looking perspective and reassure readers that the authors recognize the complexity beyond chlorophyll concentration alone.

The authors should do a careful read-through for minor language issues. For instance, in one sentence the phrase “For an example, in Andvord Bay...” is used – this should be “For example, in Andvord Bay...”. Such small grammatical errors are few, but correcting them will improve readability.

Below, I provide my specific comments.

Specific comments

I believe that a title that reflects your actual work is:

“Impact of Glacial Meltwater on Phytoplankton Chl-a along the Western Antarctic Peninsula”
or at maximum:

“Impact of Glacial Meltwater on Phytoplankton biomass along the Western Antarctic Peninsula”.

Line 8. “However” does not fit here. The two sentences can be simply linked.

I still find the use of “ecology” inappropriate in several instances. Some suggestions here and further:

Line 11. “ecology” can be replaced by “biomass”.

Line 15-17: “with an additional potential contribution from surface ocean stabilization, although the latter effect can be moderated by wind-driven mixing and sea-ice variability”. I suggest changing to e.g., “with an additional potential contribution from surface ocean stabilization due to the presence of sea ice”.

Line 17-19: This sentence is still unclear and not entirely true. I’d suggest replacing it with e.g. “Achievable phytoplankton biomass depends on the interplay between light and nutrient limitation. Our results indicate that....”

Line 30-31: Why is only the 20th-century warming reported? Could you please provide more updated information, including the past 25 years? Especially cause the data later presented are within this latter range.

Line 38: Please be sure this reference points to “abundance”, not something else.

Line 38: Replace “diatoms “ with “algae”.

Line 39,40: “role” twice, consider replacing one of them.

Line 42-43: Replace “diatom population that is released into the water column during ice melt” with “food source”. Antarctic krill can access sea-ice brines.

Line 48-49: “beyond the growing season” should be removed.

Line 53: You should add here something about which kind of “taxonomic shifts”. Please revise.

Line 68: “locales”?

Line 72: Remove “an” between “for” and “example”.

Line 74-75: You should explain in which way the phytoplankton community and biomass were found to be affected. Please, revise.

Line 75-76: Since you made a distinction between dFe in spring and fall, you should explain to what season your pFe concentration corresponds. Please, revise.

Line 76: Replace “particular” with “particulate”.

Line 92-93: should be removed as results are not presented yet.

Line 107-108. I still believe that adding data on winds would help you solve the puzzle of the MLD. I haven’t asked you

earlier to modify the current figure 2 but to add information on winds, which could be done in another panel and in relation to the other variables you have studied. Wind reanalysis are widely accessible and available and could be easily extracted for the study area and time period of interest.

Line 110: change “abundance” to “biomass”.

Line 110-111: “especially during broad-scale events captured in climatological and multi-decadal time series”, this statement is unclear, please revise.

Line 110, 116, 123: change “abundance” to “biomass”.

Line 113: change “variability” to “biomass”.

Line 117-118: I don't understand why you report on other polar regions, and the next sentence is back to the WAP. Please, explain/revise.

Line 123-125: This sentence contains a statement that should be explained or referenced.

Caption Fig.2: Change “abundance” to “Chl-a”.

Line 132: Please be sure this reference points to “abundance”, not something else.

Line 132: It is unclear what “this way” refers to. Please, revise.

Line 137, 140, 144: change “abundance” to “biomass”.

Line 151-152: This statement should be verified with actual data or referenced.

Line 167: I believe a “over” is missing between “dominate” and “the”.

Line 198: I suggest replacing “largely” with “likely”.

Line 204: This “shift in climate conditions” should be further explained. Please, revise.

Line 206: cannot be called “trends” during such a short time period, please remove it

Line 214: What are the “upper-bound uncertainty estimates”? Please revise/explain.

Line 223: “In contrast” does not fit here, I'd remove it.

Line 227: I am unsure how “remote sensing efforts” can help “dFe data”? Please, explain/revise.

Line 229: Please, remove “ecological”. This should be written as “the impact of sea-ice loss on surface Chl-a concentrations...”

Line 234-235: I suggest replacing with “ Sea ice can seed and initiate ...”

Line 280: Remove “abundance”.

Line 287, 289: Replace “abundance” with “biomass”.

Line 309-318: The description of how the sGMW is derived from satellite imagery is especially crucial. I would encourage the authors to ensure that enough of that method is summarized here (or in the supplemental material) so that readers of this paper can understand it without needing to dig deeply into the other paper.

Line 369-371: I'd move these lines within Fig. 1's caption.

Dear Reviewers,

Thank you so much for providing your thoughtful and constructive feedback on our manuscript. We greatly appreciate the time and effort you devoted to reviewing our work and offering suggestions. Your comments have helped us clarify our arguments and strengthen the overall manuscript.

In our revised submission, we have carefully addressed each of your points through:

- Point-by-point response in the attached “COMMSENV-24-3209_Response_to_Reviewers.docx” document, detailing how we integrated all recommendations into the manuscript text.
 - Our responses follow the prompt “**Response**.” under each comment.
 - Reference to L.# indicate Line numbers in the tracked changes document when all tracked changes are displayed.
 - L.# is followed by the first few words [in brackets] from the revised sentences for clarity.
- Tracked Changes in the revised manuscript to highlight exactly where and how we incorporated your feedback (COMMSENV-24-3209_Manuscript_wRevisions_2025_0228.docx).
- A clean copy of the revised manuscript, which reflects all final edits (COMMSENV-24-3209_Manuscript_CleanCopy_2025_0306.docx).

We have ensured that these revisions addressed all major and minor concerns from all three Reviewers. We look forward to any additional insights you might have.

Thank you again for your valuable input and for considering our revised manuscript for publication.

Sincerely,

B. Jack Pan

March 2025

REVIEWER COMMENTS:

Reviewer #1 (Remarks to the Author):

Here, Pan et al. present analysis linking sea ice coverage, glacial melt water, and phytoplankton biomass, finding that, as sea ice coverage in the West Antarctic Peninsula has declined since 1979, phytoplankton biomass has stayed high due to glacial melt water providing nutrients and stratifying the surface ocean. This manuscript is well written throughout and the data are well presented, and I find the conclusions of a link between phytoplankton biomass and glacial melt water convincing. However, there are two major points in the paper, both relating to MLD, that require clarification. First, the authors state that one of the primary reasons behind the correlation between phytoplankton biomass and glacial melt water concentration is that the glacial melt water helps to stratify the water column, providing more light to phytoplankton. However, their results indicate that years with high glacial melt water can have shallower or deeper MLDs. The authors argue that MLD is a transient/'snapshot in time' metric in the body of the paper, but claim in the abstract and conclusions that GMW is associated with more stable/shallower MLDs, a claim which isn't supported by their results. Looking at their results (Fig. 2b specifically), it seems to me that sea ice duration is the greatest control on MLD - years with anomalously long SI duration tend to have shallower MLDs, while years with short SI duration have deeper MLDs. Secondly, since this paper presents surface Chl concentration and not integrated primary production, slightly more care should be taken in claims of the impacts of glacial melt water on phytoplankton communities and the ecosystem. If MLD is shallower in one year, a large change in phytoplankton biomass over a thinner layer of the ocean may not correspond to any change in phytoplankton productivity (or could even coincide with a reduction in NPP).

Response: We thank the Reviewer for bringing attention to the need for clearer discussions of the following two topics:

(1) Regarding how GMW interacts with other drivers (e.g., wind forcing and sea-ice duration) to influence the MLD. While GMW freshening does promote near-surface stratification and more optimal light conditions in an instantaneous sense, this effect can be offset when wind-driven mixing events occur or when sea-ice patterns deviate from the climatology, leading to deeper MLDs despite high GMW. An additional intent here is to succinctly reflect on the importance of leveraging observations across multiple temporal scales to achieve a more holistic understanding of the interaction among sGMW, MLD, wind forcing, and sea ice. Accordingly, we have expanded our discussion in the “Glacial Meltwater & Phytoplankton” section (around L.122, 149, 166 [Sea-ice duration can...; While glacial meltwater freshening...; Although GMW generally promotes]) to highlight that interannual MLD variability also depends on sea-ice duration, which often correlates strongly with shallower/deeper MLD trends, and on transient wind events. These clarifications ensure that the multifaceted controls on MLD are clearly represented and that we acknowledge GMW’s stabilizing role without overstating its influence relative to other key drivers, such as sea-ice duration.

(2) We appreciate the reviewer’s emphasis on the distinction between surface chl-a concentration and integrated primary production (PP). We agree that relying solely on surface chl-a can obscure potential changes in water-column biomass or productivity, especially where the MLD varies significantly or a subsurface chl-a maximum may occur. Our goal was not to claim direct equivalence between surface chl-a and total (vertically integrated) PP, but rather to use satellite-derived chl-a as a proxy to identify significant spatiotemporal patterns in phytoplankton distribution.

To address this comment, we have revised the sections “Glacial Meltwater & Phytoplankton” (around L.105 [It is important to note...]) to provide more context and purpose of relying on surface chl-a concentration in this study, as well as citing studies that indicate WAP’s hydrography often allows surface chl-a to be an effective first-order indicator of phytoplankton abundance. A few sentences are added to the Section “Glacial Meltwater Climatology” (around L.249 [Moreover, while our analysis...]) to highlight that dedicated measurements of depth-resolved phytoplankton biomass and in-situ PP, especially over the entire growing season, remain an important topic for future work. The “Methods, Data Description” on chl-a concentration has also been amended around L.279 [Throughout this study, we use...].

Minor comments:

Line 25-26: These productivity rates (from Vernet et al., 2008) are averaged over the summer/observation period, not the year. I believe the second paper cited just references the first paper cited and does not provide measured NPP.

Response: The sentence has been amended to specify that the chl-a averages describe the summer months.

Fig 2b: The Chl-a anomaly legend is difficult to understand. Consider showing the full size of the dots (not just horizontally but also vertically) and eliminating the darker grey circle (I think this is intended to show how much larger each circle is from the previous one, but I find it confusing). I'm also confused with this plot/caption - how is it that only one of the 17 years represented here has a glacial melt water concentration anomaly below the 2002-2022 climatology?

Response: We have revised Figure 2 – more specifically, we have changed the size and style of the legend in Fig. 2b according to the Reviewer’s advice. Regarding the Reviewer’s comment on glacial meltwater anomalies – the values presented in Fig. 2b represent summer surface glacial meltwater anomalies calculated based on the following (also presented in section “Data Processing & Analysis”):

$$\frac{(\text{summer mean} - \text{climatological mean})}{\text{climatological mean}} \times 100\%$$

The summer mean values reflect D/J/F of each year (described in caption) while climatological mean encompasses all available MODIS-A data which usually covers from late September to early April. Consequentially, as expected, this indicates that summer months would usually have more glacial meltwater at the sea surface in comparison to the rest of the year. These results also serve as an additional QA/QC measure to validate our results.

Lines 70-72: High Chl concentration and long sea ice duration can co-occur, but they certainly do not look correlated in Fig 2b - and thus I don't think these data convince me that 'longer sea ice duration...can lead to high Chl'. While sea ice presence can prevent wind mixing and ice melt can provide nutrients and stabilize the ocean surface, a shorter sea ice duration can also alleviate light limitation, making the relationship between sea ice and productivity more complicated than described here (but more in line with the data presented, whereby shorter sea ice duration can coincide with higher Chl).

Response: We appreciate the reviewer’s perspective on how sea ice influences surface chlorophyll (chl-a) concentration. We agree that sea-ice impacts on phytoplankton are multifaceted. Although several studies¹⁻⁴ demonstrate that a longer sea-ice duration can support higher chl-a via enhanced winter diatom seed stocks and post-melt stabilization, extensive or late-retreating ice can also limit light availability. Conversely, shorter sea-ice duration may alleviate light limitation earlier in the growing season⁵, yet it can also restrict under-ice diatom growth and nutrient retention. Thus, as shown in Fig. 2b, the correlation between sea-ice duration and chl-a is complex and depends strongly on the timing and nature of ice advance and retreat. In response to the reviewer’s comment, we have revised the text around L.166 [Although sGMW generally...].

Lines 86-88: I think here the authors again note a co-occurrence and subsequently (in the abstract and concluding paragraph) conflate that co-occurrence with causality. Looking at Fig 2b, I would argue that it appears that sea ice duration is the greatest control on MLD, not GMW, but that GMW is a greater control on Chl concentration than sea ice concentration.

Response: Thank you for pointing out the potential conflation of correlation with causality in our discussion of GMW and MLD. We agree that sea-ice duration is an important driver of MLD, often complicating and even overshadowing GMW’s stabilizing effect on seasonal time scales. At the same time, as mentioned by the Reviewer, our data indicate that GMW tends to be more

strongly correlated with changes in chl-a concentration – presumably via nutrient supply and localized freshening – than with major shifts in MLD. In order to reflect these comments from the Reviewer, we have revised text around L.117 [Over an annual timescale...].

Line 90: I would replace the end of this sentence with "offshore onto the wider continental shelf."

Response: The sentence now reads: "Prolonged inshore observations have found that wind events, particularly katabatic winds, are an important mechanism for transporting glacial meltwater away from fjords' glacio-marine interfaces and towards offshore onto the wider continental shelf"

Line 99: I don't like the use of 'coupled' here and would replace with correlated, for example.

Response: The text has been revised accordingly – the word "coupled" has been replaced with "correlated."

Line 129: A sentence discussing why these time periods (2002-2018 and 2019-2021) were chosen for analysis would be helpful.

Response: A sentence has been added to section "Glacial Meltwater Climatology" around L.204 [2019-2021 is based on...] to reflect the reasoning behind why these time periods were chosen. Additionally, the last section of the Methods section also provides some additional details on this topic.

Line 137: I would love to see these trends quantified in $d\ yr^{-1}$, especially since it's hard to compare them in Figure 5 due to the different scales of the y axes.

Response: Thank you for your suggestion to quantify the sea-ice trends in $d\ yr^{-1}$. We agree that such a metric can be very insightful for comparing multiple time series on a common scale. However, detailed trend analyses of this kind extend beyond the scope of our current study, which focuses primarily on the relationship between interannual sea-ice variability, glacial meltwater, and phytoplankton. Instead, we refer to Stammerjohn et al. (2008)⁶ for in-depth calculations of WAP sea-ice trends over earlier decades and to Reid et al. (2024)⁷ for updated analyses. These references provide the quantitative detail the Reviewer suggested, including more robust trend estimates in $d\ yr^{-1}$ for the WAP region.

Lines 144-147: I believe the first part of the first sentence here is referring to sea ice algae. While there can be very high concentrations of SIA in the Southern Ocean, they are pretty unimportant in terms of total NPP. They are critical in terms of timing of the availability of primary production, but that's not really discussed elsewhere in this paper. This sentence and the next appear to conflate SIA and phytoplankton, so I might rephrase/omit to avoid confusion.

Response: We agree that SIA is ecologically significant for the timing of primary production. In our revision, we have revised around L.234 to reflect this comment [A main ecological significance...].

Line 164: Again, I'm not certain that your analysis demonstrates that glacial melt water stabilizes the surface ocean.

Response: The text has been omitted to reflect this change in L.262 [~~and stabilizing...~~].

=====

To Reviewer #1: Thank you again for your thoughtful and constructive comments. Your observations and suggestions were instrumental in helping us refine our discussion of glacial meltwater's role in the water column and in clarifying the relationship between surface chl-a and integrated PP and phytoplankton biomass. We appreciate the time and effort you devoted to reviewing our work and hope our revisions addressed your concerns.

Reviewer #2 (Remarks to the Author):

The study of Pan et al. examines the ecological impact of glacial meltwater on phytoplankton along the Western Antarctic Peninsula, using 20 years of multi-source data. The results reveal a correlation between meltwater and phytoplankton biomass, that is suggested to be driven by nutrient fertilization and surface ocean stabilization. The findings suggest that increased glacial meltwater may mitigate the negative effects of sea-ice loss on phytoplankton productivity, emphasizing the importance of meltwater as a critical driver in polar coastal ecosystems.

The results, based on Pan et al. previous work using remote sensing for estimating surface glacial melt water, are quite convincing and certainly relevant for a broad research community studying the impacts of global change in polar regions. However, the complexity of ice-ocean-ecosystem interactions requires more thorough conceptual exposition. While the discussion touches on these aspects, they are not adequately framed earlier in the manuscript, leaving the discussion somewhat fragmented.

My suggestion is to (in the introduction) describe the system better than it is currently with specific attention to:

The sea ice: Elaboration on the process "seeding" (Line 36-38) and the foundational role of sea-ice algae in initiating blooms by providing the initial population of diatoms and contributing to the conditions needed for their proliferation – including their role as Fe and other nutrient 'keepers'.

Response: We thank the Reviewer for emphasizing the importance of sea ice and sea-ice algae (SIA) in setting the stage for phytoplankton blooms. In response, we have revised the Introduction (around L.39 [Sea ice not only serves as a physical barrier...]) to provide a more detailed description of sea ice algae, nutrient and iron retention, and contextualizing SIA in the broader bloom cycle.

The glaciological setting of glaciers in the WAP and the mechanisms involved in their regulation of oceanic nutrients: This is critical to understanding their impact on primary productivity. A key question is which glacier types—marine-terminating or land-terminating—are most relevant in this region. The mechanisms through which these glacier types deliver meltwater to the ocean differ significantly and have direct implications for nutrient content. For marine terminating WAP glaciers, what is the partitioning between surface melting, calving, and submarine melting? This needs to be introduced and explained to provide context, as it is important for linking meltwater production to ocean fertilization processes. Specifically, the surface melt, and hence subglacial discharge, is important to understand for WAP, as it can influence phytoplankton growth by:

1. Upwelling of nitrate-rich waters: Subglacial discharge promotes upwelling, bringing deep, nutrient-laden waters into the photic zone, where phytoplankton can use it.
2. Sediment transport: Subglacial processes determine the extent and composition of sediments transported to the ocean, impacting nearshore and offshore ecosystems.

These mechanisms should be briefly explained. I am sure the authors are well aware of all of this, but it does not come out clearly in the current manuscript.

Response: We appreciate the Reviewer's advice on clarifying the context of the WAP glaciology and sediment transport. In the revised manuscript, we have added a dedicated paragraph in the Introduction (around L.54 [Glaciers along the WAP exhibit...]) describing:

(1) Types of Glaciers in the WAP where we note that most glaciers on the western side of the Antarctic Peninsula are marine-terminating (e.g., tidewater glaciers), whereas some land-terminating glaciers also exist in localized regions. This distinction matters because tidewater glaciers can experience submarine melting⁸ and subglacial discharge, while land-terminating glaciers melt primarily via surface runoff and calving.

(2) Mechanisms of Meltwater Production and Nutrient Flux

- Subglacial and Submarine Melting: Subglacial discharge can promote upwelling of deep, nitrate-rich waters into the photic zone⁹. Submarine melting also releases meltwater at depth, which can entrain or mix with Circumpolar Deep Water (CDW).
- Surface Melting and Calving: While calving contributes freshwater inputs and icebergs, surface melt typically drains internally or along the glacier front, carrying sediments and

nutrients into nearshore waters.

- Sediment Transport and Biogeochemical Implications: Glacial melt can entrain sediments, potentially affecting light availability and nutrient supply^{10,11}. We briefly addressed how these processes, though important, vary spatially depending on glacier geometry and bed topography.

Another question in this regard is also relevant to address: To what extent do WAP glaciers produce sediment? It is likely limited, given historically low surface melt in the region, which might not have primed glacier beds for substantial soft sediment production. Furthermore, the bioavailability of these sediments for primary producers remains an open question. While the discussion suggests sediment increases in the future could limit sunlight in coastal systems, it is unclear how this might impact offshore ecosystems.

Response: We appreciate the query regarding sediment production by WAP glaciers and sediments' bioavailability. Our response addresses both the current extent of sediment delivery and the future implications for nearshore vs. offshore ecosystems. Accordingly, we have revised the text around L.66 [Glacial meltwater can entrain both...]. The following content was added:

(1) Current sediment production in the WAP: while many WAP glaciers still exhibit relatively low melt rates, local “hotspots” exist (e.g., Potter Cove) where retreating tidewater glaciers discharge significant sediment loads^{12,13}. These cases illustrate how intense sediment outflow can reduce water clarity and affect phytoplankton ecology in nearshore environments.

(2) Sediment bioavailability and nutrient dynamics: we have references such as studies by Forsch et al. (2021)¹⁴ demonstrating that sediment plumes can be a source of iron and other micronutrients, potentially benefiting phytoplankton growth. However, the fraction of sediments that is truly bioavailable (vs. particles that simply attenuate light) remains an active area of research and likely varies by glacier type and bedrock composition.

(3) Nearshore vs. offshore impact: In response to the Reviewer's remark about offshore ecosystems, we now note that while sediment plumes can be diluted or dispersed offshore, the long-range effects are typically more muted than in the vicinity of glacial fronts or near the fjords. Nonetheless, strong wind events and currents occasionally transport these turbid waters beyond the coastal zone^{13,15}, highlighting the potential for episodic offshore impacts under future scenarios of increased melt.

Although the authors cannot resolve all these unknowns, they should describe these system complexities better in the introduction. This will allow their findings—i.e. the observed increase in phytoplankton growth despite low sea-ice years alongside higher glacial meltwater input—to fit more effectively into the broader puzzle of WAP ecosystem dynamics. By framing their study within a better outline of the understanding of the WAP glaciomarine environment, they can enhance the impact of their specific data and conclusions and provide recommendations for future studies in the region.

Response: Thank you for this valuable suggestion. We have expanded the Introduction to provide a more comprehensive overview of the WAP glaciomarine environment, including: (1) sea ice influences on phytoplankton bloom dynamics around L.39 (Sea ice not only serves...). (2) Glacial setting of the WAP, focusing on different types of glaciers and their associated meltwater characteristics (surface melt, subglacial discharge, submarine melting etc.) can contribute to nutrient fluxes, around L.54 (Glaciers along the WAP exhibit...). (3) Potential sediment impacts from tidewater glaciers, emphasizing both the light-attenuation aspect and the delivery iron or other nutrients, around L.69 (These sediments may negatively impact...).

Some additional comments:

Fig. 2 Add details such as the time period and data type directly to Figure 2, along with clear color references. For MODIS data, include references to methodologies used to obtain these estimates.

Response: Fig. 2 has been updated to include time period, data type, and reference to methodologies (in addition to existing information in figure caption and Methods section).

Line 44-49: The previous work of the main author in the region is outlined (Pan et al. Environmental drivers of phytoplankton taxonomic composition in an 357 Antarctic fjord. Prog. Oceanogr. 183, 102295 (2020) and Pan, et al. Remote sensing of sea surface glacial meltwater on the Antarctic Peninsula shelf. 1–16 (2023) doi:10.3389/fmars.2023.1209159.). I assume that the novelty of the study here refers to the longer time period under study.

However, I cannot confirm this without reviewing what time periods the two earlier papers by the author covers. The novelty could easily be clarified with more precise wording in line 44-47, and line 49.

Response: The text around L.86 [Prior studies on this subject] now reflect this change by expanding in time and space and highlight the novelty of this study in comparison to prior work.

Line 59: Revise the section header to better reflect its content, e.g., 'Glacial meltwater drives offshore phytoplankton growth'.

Response: The section header has been revised accordingly.

Line 81: The authors could start this section by very briefly explaining (1-2 sentences) the relevance of the MLD for the primary producers (i.e. regulating nutrient and light availability etc). It will provide better context for the further discussion in the same section (line 81-96).

Response: The text around L.141 [MLD strongly influences phytoplankton] has been amended to provide the suggested brief overview and explanation.

Line 131: To add clarity provide reference to Fig 4 for this statement.

Response: The text has been updated to include a reference to Fig. 4.

Line 134-136: The uncertainties on the sGMW fractions are quite low, and there is no mentioning in the manuscript how the uncertainties are assessed. If they were higher, any robustness of the observed increase (2.52% to 2.82%) would be less convincing.

Response: We appreciate the reviewer's concern about uncertainty quantification. To address this comment, we have added a sentence in the Methods section clarifying how the uncertainties in sGMW fraction were derived (in the sGMW section of Methods). Specifically, they likely stem from the propagation of uncertainties in remote sensing reflectance, modeled glacial meltwater end-members, and stable isotope calibration. We also referenced these uncertainty bounds and their calculation in our revised discussion as well, noting that even at the upper bound of uncertainty, the difference remains statistically significant. The text has been updated around L.214, 313 [the observed increase...; The mean sGMW fraction...]

In the method section: Spell out abbreviations and provide error estimates for the key metrics.

Response: The text in the Methods section has been updated to include the full data variable names instead of abbreviations.

=====

To Reviewer #2: We appreciate your detailed and thoughtful feedback, particularly regarding glacier types, sediment transport, and the role of sea-ice algae. Your suggestions helped us expand the background information and provide more comprehensive context on these processes, enhancing the manuscript's clarity and impact. Thank you for your time and insights.

Reviewer #3 (Remarks to the Author):

General comments

The manuscript by Pan et al. examines the influence of sea surface glacial meltwater (sGMW) on chlorophyll-a (Chl-a) dynamics along the Western Antarctic Peninsula (WAP). Utilizing a dataset spanning two decades, the study suggests that glacial meltwater influences phytoplankton biomass through nutrient enrichment, surface stabilization, and light optimization. The findings propose that sGMW mitigates the adverse effects of sea ice loss on primary production, offering insights into polar ecosystem resilience under climate change.

The study addresses a timely aspect of climate change in polar regions, providing insights into the role of glacial meltwater as a biogeochemical driver in a rapidly changing ecosystem. It effectively combines remote sensing, reanalysis products, and in-situ observations and is well-written.

However, several concerns need to be addressed:

-Scope of Phytoplankton Ecology: The manuscript promises to focus on impacts on phytoplankton ecology; however, the only biogeochemical variable explored is chlorophyll-a. There are references to phytoplankton biomass, but no explicit biomass data are reported. Changes in Chl-a concentration can also result from shifts in community composition, which are not discussed.

Response: We appreciate the Reviewer's concern that our manuscript focuses predominantly on remotely sensed chl-a as a proxy for phytoplankton biomass, rather than directly measuring species composition or other biogeochemical variables. Our overarching goal is to capture broad spatial and temporal patterns in the WAP region over two decades, and chl-a from remote sensing offers a high-resolution time series for this purpose; detailed species-level dynamics, while critical for a fuller ecological understanding, are beyond the immediate scope of this study. Nevertheless, we acknowledge that chl-a alone cannot distinguish taxonomic shifts or detailed community composition changes. In response, we have revised text around L.105 [It is important to note...] as well as amended the text for chl-a measurement in the Methods section.

-Nutrient Fertilization Attribution: The attribution of changes to nutrient fertilization, particularly iron, is plausible but not substantiated with direct evidence. Furthermore, if sea ice serves as an iron source, its decline would also reduce its fertilization effect.

Response: Thank you for your concerns regarding the attribution of phytoplankton changes to iron fertilization from glacial meltwater. While our manuscript does not present direct iron measurements, multiple prior studies have demonstrated that glaciers and meltwater plumes are indeed significant sources of iron in the coastal WAP. For instance, Forsch et al. (2021)¹⁴ identified seasonal dispersal of fjord meltwaters as an important driver for dissolved iron and manganese, and Annett et al. (2015, 2017)^{16,17} similarly highlighted how subglacial discharge and meltwater inputs can contribute to elevated iron in Ryder Bay. Together, these findings bolster our argument that glacial meltwater can provide essential micronutrients to phytoplankton community.

Regarding the potential iron source from sea ice, we agree that declining sea ice may reduce Fe input from this pathway. However, the net effect on phytoplankton biomass appears mitigated or even offset by the increasing meltwater input, where sea ice likely plays a more significant role during early-season phytoplankton blooms, while glacial meltwater input becomes increasingly important later in the growing season, particularly in late summer and fall. We have expanded the discussion (around L.219 [Our results are consistent with...]) to briefly cite these key studies and clarify that while our results are consistent with iron fertilization by glacial meltwater, comprehensive iron measurements would further substantiate this mechanism. As noted, future research, including remote sensing approaches under development, will be essential for directly quantifying iron fluxes.

-Novelty of Findings: Lines 43-44 indicate that sGMW has already been identified as a driver for phytoplankton productivity in the WAP region, potentially limiting the novelty of this work.

Response: We appreciate and acknowledge the Reviewer's concern regarding novelty. While prior studies¹⁸ have indeed noted local-scale relationships between sGMW and phytoplankton, our current work substantially extends both temporal and spatial coverage in order to assess how representative this process is for the region. Specifically, (1) we analyze nearly two decades (2002–2022) of multi-platform data – including satellite-derived sGMW fractions, reanalysis products, and in-situ observations, thus enabling a broader understanding of how sGMW influences phytoplankton across different environmental regimes. (2) Quantitatively compared the relative roles of sGMW and sea ice in driving phytoplankton variability on regional scales, rather than focusing on a single fjord or short timeframe. (3) Demonstrated that sGMW has become increasingly important in recent years, potentially mitigating negative effects of declining sea ice on phytoplankton abundance over the broader WAP. (4) Utilized the novel ocean-color-based sGMW algorithm in the context of enhancing long-term ecological study; while prior work¹⁹ on this subject only focused on methodology development.

-Geographical Focus: The manuscript exclusively focuses on the WAP. A broader discussion on whether these findings can be extrapolated to other polar regions would enhance the study's relevance.

Response: Thank you for this suggestion. We have added a short statement in the discussion (L.240 [Although the WAP presents...]) noting that while the WAP has unique glaciological and oceanographic conditions, certain processes, such as meltwater-driven stratification and nutrient input, also occur in other polar regions (e.g., parts of Greenland). We now clarify these parallels and outline potential differences in climate forcing, sediment dynamics, or glacier type that may limit direct extrapolation.

Overall, while the topic is interesting and timely, additional data and analyses are necessary to strengthen the work. I wonder if these would be available from the Palmer LTER transects?

Response: We appreciate the reviewer's suggestion. The Palmer LTER program has certainly generated an extensive dataset beyond the scope of our current manuscript. We have used selected components of these data to inform and validate our sGMW and phytoplankton findings; however, a more comprehensive analysis (including taxonomic composition, nutrient fluxes, and additional interannual comparisons) is presented in a separate manuscript in preparation (Pan et al. 2024. "Environmental Drivers Modulate Phytoplankton Community Seasonal Succession in the Western Antarctic Peninsula."). In that work, we leveraged over 20 years of in-situ CTD and HPLC pigment data from the Palmer LTER to examine how sea-ice and glacial meltwater dynamics shape phytoplankton community seasonal succession across coastal and shelf zones. Because our current study aims to synthesize broader spatiotemporal patterns using data aggregation approaches, we have kept our use of additional Palmer LTER data focused on validating key relationships (e.g., meltwater vs. chl-a relationships).

Specific Comments

Title: The term "phytoplankton ecology" is not adequately addressed in the manuscript. Consider revising the title to reflect the study's actual focus.

Response: Thank you very much for this suggestion. We appreciate your perspective on the scope implied by "phytoplankton ecology." While our manuscript focuses primarily on spatiotemporal patterns of phytoplankton abundance (as inferred from chl-a) and the environmental drivers behind those patterns, we view this as a meaningful examination of the ecological processes. For instance, how meltwater fertilization, sea-ice variability, and upper ocean mixing collectively shape phytoplankton growth and distribution. Therefore, we believe our current title conveys the study's central aim of connecting glacial meltwater with phytoplankton ecological dynamics in the region. Accordingly, we prefer to keep the existing title. However, we have added clarifications throughout the text to emphasize what aspects of "phytoplankton ecology" we are addressing and to acknowledge those that lie outside our scope. We hope these revisions further highlights why phytoplankton ecology remains integral to our study's focus.

Abstract: Line 13: Replace "phytoplankton biomass" with "Chl-a" to accurately represent the measured variable.

Response: The Abstract has been revised accordingly.

Main Text:

Line 24: Consider removing the word "polar" for conciseness.

Response: The Abstract has been revised accordingly.

Lines 38-39: Expand on the concept of ice algae seeding to provide clarity.

Response: The text has been revised accordingly.

Line 41: Rephrase "Sea ice ...hosting under-ice diatoms..." for clarity, as sea ice hosting under-ice species is contradictory.

Response: The text has been revised accordingly. It now reads: "Sea ice also offers a habitat beneath its underside, where diatoms can thrive, and it plays a critical role in the life cycle of Antarctic krill (*Euphausia superba*)."

Lines 46-47: Elaborate on these findings, particularly regarding community composition.

Response: Thank you for your suggestion. While we recognize that the specific community composition offers valuable insights, our study primarily focuses on assessing how sea surface sGMW influences overall phytoplankton abundance. Although important, detailed taxonomic changes or community composition analyses lie beyond the scope of this study. Nonetheless, we have added a brief statement (L.52 [While these prior work...]) referencing relevant studies that address taxonomic composition in more depth and clarifying the limits of our own focus here.

Lines 54-56: Revise this section, as the findings do not clearly support the stated "ecological impact."

Response: Thanks for this suggestion. This entire section has been revised to include discussions regarding the ecological impact on phytoplankton in respect to glacial meltwater processes, nutrient fluxes, sediment loading associated with meltwater etc. (starting in L.52 [While prior work documented taxonomic shifts...]). Other revised sections further elaborate on how these processes can have an ecological impact.

Lines 68-69: Clarify how these observations enhance understanding of sGMW's impact on marine ecology.

Response: Thanks for this suggestion. This sentence has been revised to “By examining these observations on an annual timescale, we track how sGMW affects water-column processes and phytoplankton variability both throughout the growing season and from year to year, thereby enhancing our understanding of its ecological role.” Moreover, the entire paragraph has also been revised to include a more detailed discussion of our results.

Lines 70-73: Expand on the references provided, as the statement in lines 70-72 is not universally applicable.

Response: Thanks for this suggestion. This statement has been revised accordingly around L.130 [This result is consistent with...].

Lines 77-78: Acknowledge that Chl-a concentrations are not always indicative of phytoplankton abundance; please revise.

Response: A statement has been added to address comment accordingly around L.116 [Although we use chl-a as a proxy...].

Line 79: Replace the word "ecological" with a more appropriate term.

Response: Text has been revised to clarify that this is regarding “a coherent, region-wide ecological impact of sGMW on phytoplankton abundance across the entire WAP region.”

Lines 93-94: An analysis of wind patterns would enhance the robustness of the study; consider incorporating this aspect.

Response: Text has been amended to include discussion on wind forcing around L.121 [wind forcing often exert...]

Line 107: This statement is self-evident; consider rephrasing or removing.

Response: This sentence has been rephased to “Together, elevated SST and SAT drive glacial meltwater production over the course of each year, with the strongest effects occurring during the summer (Fig. 3d, 3h)” (around L.180).

Line 125: Clarify what is meant by "unfavourable light conditions."

Response: This sentence has been revised to read: “This decoupling is largely driven by declining sunlight in late summer (shorter daylength and lower solar angle), which reduces light availability for phytoplankton growth.”

Line 132: Using the mean may not be informative; consider analyzing seasonal maxima in relation to ice season duration and sGMW for more insightful information.

Response: Thank you for this thoughtful recommendation. We agree that examining seasonal maxima could offer additional nuances. However, our primary goal here is to characterize broad spatiotemporal trends across nearly two decades rather than pinpoint specific peak events. Averaging the data is more aligned with that objective, providing a synoptic measure of annual or seasonal-scale variability rather than episodic extremes. Furthermore, our existing dataset and analysis framework are optimized for interpreting mean conditions. In our aforementioned ongoing work, particularly with a higher temporal resolution dataset (i.e., seasonal), we are exploring the insights gained from comparing seasonal maxima to sGMW and sea ice duration.

Line 134: If the result is significant, report the p-value.

Response: p-values and other detailed statistical analyses are reported in the “Data Processing & Analysis” section of Methods.

Lines 142-144: Provide further evidence or explore alternative explanations for the stated results.

Response: This statement has been amended, in addition to other added explanations throughout earlier text to explore alternative explanation for these results.

Lines 151-153: This point is intriguing and warrants further development.

Response: A statement is added to the end of this paragraph to further develop this point [As sediment-laden meltwater becomes more common in the WAP...].

Lines 166-170: Expand this section to provide more depth.

Response: This section has been revised to provide more context.

Line 181: Phytoplankton abundance is not depicted in Fig. 2a; please revise accordingly.

Response: We appreciate the reviewer's observation regarding the term "phytoplankton abundance." In our manuscript, chl-a concentration is used as a widely accepted proxy for phytoplankton abundance²⁰. While chl-a does not directly measure cell counts or species composition and there are many nuanced studies regarding the relationship between chl-a and phytoplankton abundance, it generally and reliably reflects relative changes in phytoplankton biomass at regional scales^{21–24}. Therefore, we respectfully choose to retain the current figure and terminology, but we have added a clarifying note throughout the text, indicating that chl-a is being treated as a proxy for abundance rather than a literal cell count.

Figures:

Figure 1: Including additional information on currents and winds would enhance the figure's informative value.

Response: Thank you for suggesting additional wind and current data in Figure 1. While we agree that these factors are important, our figure is already at the threshold of visual complexity. We wish to keep it focused on the primary features – namely, the distribution of glacial meltwater along the WAP and key sampling locations. Adding wind and current vectors could overwhelm the figure and reduce clarity. Instead, we have added additional text throughout the main text that details the impact of winds and currents in the region. We hope this approach maintains the figure's clarity while ensuring further information is presented on local circulation patterns.

Figure 2: There is no data related to "phytoplankton abundance" presented; please revise the figure or its description.

Response: Please see our comment above regarding "Line 181: Phytoplankton abundance is not depicted in Fig. 2a; please revise accordingly."

Grammar and Clarity Corrections

Replace terms like "ecological" with more precise language relevant to the context.

Response: Thank you for noting that precision in terminology is crucial. We have, where possible, substituted "ecological" with more specific terms when referencing narrower processes. However, in certain places we continue using "ecological" or "ecology" because they encompass the broader interactions between organisms and both their abiotic and biotic environments – a core concept in our discussion of how glacial meltwater influences phytoplankton within the WAP ecosystem. In these specific instances, we believe that retaining "ecological" in this broader sense remains appropriate and clear for the scope of our study.

Clarify statements that are currently vague or self-evident to enhance readability and comprehension.

Response: Thank you for highlighting this concern. We conducted a careful review of the manuscript and revised or expanded the text for better precision and readability. We believe these modifications based on your recommendations have improved the overall clarity and coherence of our article.

=====

To Reviewer #3: Thank you for your valuable comments and recommendations. Your

suggestions prompted important clarifications, particularly regarding terminology precision, the scope of our study, and the broader relevance of our findings. We greatly appreciate your guidance in helping improve the manuscript's clarity and scientific rigor

References

1. Garibotti, I. a., Vernet, M., Kozłowski, W. a. & Ferrario, M. E. Composition and biomass of phytoplankton assemblages in coastal Antarctic waters: A comparison of chemotaxonomic and microscopic analyses. *Mar. Ecol. Prog. Ser.* **247**, 27–42 (2003).
2. Garibotti, I. A., Vernet, M. & Ferrario, M. E. Annually recurrent phytoplanktonic assemblages during summer in the seasonal ice zone west of the Antarctic Peninsula (Southern Ocean). *Deep. Res. Part I Oceanogr. Res. Pap.* **52**, 1823–1841 (2005).
3. Vernet, M. *et al.* Primary production within the sea-ice zone west of the Antarctic Peninsula: I—Sea ice, summer mixed layer, and irradiance. *Deep Sea Res. Part II Top. Stud. Oceanogr.* **55**, 2068–2085 (2008).
4. Vernet, M. *et al.* Primary production within the sea-ice zone west of the Antarctic Peninsula: I—Sea ice, summer mixed layer, and irradiance. *Deep Sea Res. Part II Top. Stud. Oceanogr.* **55**, 2068–2085 (2008).
5. Moreau, S. *et al.* Climate change enhances primary production in the western Antarctic Peninsula. *Glob. Chang. Biol.* **21**, 2191–2205 (2015).
6. Stammerjohn, S. E., Martinson, D. G., Smith, R. C., Yuan, X. & Rind, D. Trends in Antarctic annual sea ice retreat and advance and their relation to El Niño–Southern Oscillation and Southern Annular Mode variability. *J. Geophys. Res.* **113**, 1–20 (2008).
7. Clem, K. R. *et al.* STATE OF THE CLIMATE IN 2021 ANTARCTICA AND THE SOUTHERN OCEAN STATE OF THE CLIMATE IN 2021 Antarctica and the Southern Ocean Editors. **103**, 307–340 (2022).
8. Cook, A. J. *et al.* Ocean forcing of glacier retreat in the western Antarctic Peninsula. *Science (80-.)*. **353**, 283–286 (2016).
9. Cape, M. R. *et al.* Circumpolar Deep Water impacts glacial meltwater export and coastal biogeochemical cycling along the west Antarctic Peninsula. *Front. Mar. Sci.* **6**, 144 (2019).
10. Domack, E. W. & Ishman, S. Oceanographic and physiographic controls on modern sedimentation within Antarctic fjords. *Geol. Soc. Am. Bull.* **105**, 1175–1189 (1993).
11. Eidam, E. F., Nittrouer, C. A., Lundesgaard, Homolka, K. K. & Smith, C. R. Variability of Sediment Accumulation Rates in an Antarctic Fjord. *Geophys. Res. Lett.* **46**, 13271–13280 (2019).
12. Schloss, I. R. *et al.* Response of phytoplankton dynamics to 19-year (1991-2009) climate trends in Potter Cove (Antarctica). *J. Mar. Syst.* **92**, 53–66 (2012).
13. Pan, B. J., Vernet, M., Reynolds, R. A., Greg Mitchell, B. & Mitchell, B. G. The optical and biological properties of glacial meltwater in an Antarctic fjord. *PLoS One* **14**, 1–30 (2019).

14. Forsch, K. *et al.* Seasonal dispersal of fjord meltwaters as an important source of iron to coastal Antarctic phytoplankton. *Biogeosciences Discuss.* 1–49 (2021) doi:10.5194/bg-2021-79.
15. Lundesgaard, Ø. *et al.* Hydrography and energetics of a cold subpolar fjord: Andvord Bay, western Antarctic Peninsula. *Prog. Oceanogr.* **181**, 102224 (2020).
16. Annett, A. L. *et al.* Comparative roles of upwelling and glacial iron sources in Ryder Bay, coastal western Antarctic Peninsula. *Mar. Chem.* **176**, 21–33 (2015).
17. Annett, A. L. *et al.* Controls on dissolved and particulate iron distributions in surface waters of the Western Antarctic Peninsula shelf. *Mar. Chem.* (2017) doi:10.1016/j.marchem.2017.06.004.
18. Pan, B. J. *et al.* Environmental drivers of phytoplankton taxonomic composition in an Antarctic fjord. *Prog. Oceanogr.* **183**, 102295 (2020).
19. Pan, B. J. *et al.* Remote sensing of sea surface glacial meltwater on the Antarctic Peninsula shelf. 1–16 (2023) doi:10.3389/fmars.2023.1209159.
20. Behrenfeld, M. J. & Falkowski, P. G. Photosynthetic rates derived from satellite-based chlorophyll concentration. *Limnol. Oceanogr.* **42**, 1–20 (1997).
21. Sauzède, R. *et al.* Vertical distribution of chlorophyll a concentration and phytoplankton community composition from in situ fluorescence profiles: A first database for the global ocean. *Earth Syst. Sci. Data* **7**, 261–273 (2015).
22. Al Diana, N. Z., Sari, L. A., Arsad, S., Pursetyo, K. T. & Cahyoko, Y. Monitoring of Phytoplankton Abundance and Chlorophyll-a Content in the Estuary of Banjar Kemuning River, Sidoarjo Regency, East Java. *J. Ecol. Eng.* **22**, 29–35 (2020).
23. Bock, N., Subramaniam, A., Juhl, A. R., Montoya, J. & Duhamel, S. Quantifying Per-Cell Chlorophyll a in Natural Picophytoplankton Populations Using Fluorescence-Activated Cell Sorting. *Front. Mar. Sci.* **9**, 1–12 (2022).
24. Petit, F. *et al.* Influence of the phytoplankton community composition on the in situ fluorescence signal: Implication for an improved estimation of the chlorophyll-a concentration from BioGeoChemical-Argo profiling floats. *Front. Mar. Sci.* **9**, 1–16 (2022).

Dear Reviewers,

Thank you so much for providing additional feedback on our manuscript. We once again appreciate the time and effort you devoted to reviewing our work and offering suggestions. Your comments have helped us clarify our arguments and strengthen the overall manuscript.

In our revised submission, we have carefully addressed each of your points through:

- Point-by-point response in this document, detailing how we integrated all recommendations into the manuscript text.
 - Our responses follow the prompt “**Response:**” under each comment.
 - Reference to L.# indicate Line numbers in the tracked changes document when all tracked changes are displayed (Main_COMMSENV-24-3209_Manuscript_wRevisions_2025_0423.pdf).
- Tracked Changes in the revised manuscript to highlight exactly where and how we incorporated your feedback (Main_COMMSENV-24-3209_Manuscript_wRevisions_2025_0423.pdf).
- A clean copy of the revised manuscript, which reflects all final edits (Main_COMMSENV-24-3209_Manuscript_CleanCopy_2025_0423.pdf).

We have ensured that these revisions addressed all major and specific concerns.

Thank you again for your valuable input and for considering our revised manuscript for publication.

Sincerely,

B. Jack Pan

May 2025

REVIEWERS' COMMENTS:

Reviewer #1 (Remarks to the Author):

The authors have satisfied my major concerns with their edits to this work, and I appreciate their efforts to revise the manuscript. However, I don't understand why reporting the slopes (in days per year) of the linear regressions they present in Fig. 5 is "beyond the scope of this work." Including only the number of sea ice covered days in 1979 and 2022 gives readers a very skewed sense of the linear regression of the timeseries, which shows a far more moderate trend. I would also revise L124 - if the aim is to talk about the potential that sea ice algae seed phytoplankton blooms, phrasing such as "facilitates the accumulation of sea ice algae that may seed phytoplankton blooms" would be clearer.

Response: Thank you very much for your comments! I had likely misunderstood your previous comment regarding the slope in the figure. The slopes (m) are now reported on page 8 (L.214 to 216 in Section "Glacial Meltwater Climatology").

We have also revised the manuscript accordingly in L.120 to 121 on page 4 (previous L.124 in the copy that included all revisions).

Reviewer #2 (Remarks to the Author):

The authors have done a very thorough job in complying with the reviewers queries, concerns and suggestions, and the manuscripts i markedly improved. I have no further comments.

Response: Thank you again for your time and your previous comments.

Reviewer #3 (Remarks to the Author):

Second-Round Review of "Impact of Glacial Meltwater on Phytoplankton Ecology along the Western Antarctic Peninsula"

Please note that line numbers refer to the marked-up revised ms.

General comments

Overall, the revised manuscript has improved and addresses some key concerns raised in the first-round review. The authors have clarified their use of chlorophyll-a as a proxy for biomass (but not for abundance, see my comment below), added supporting evidence for the proposed nutrient fertilization mechanism, and expanded the discussion to place their findings in a broader polar context. The data analysis and the conclusions, while based largely on correlations, are more convincing given the additional literature support. However, I still think that providing additional data directly in the ms would make it stronger (e.g., winds, see my recommendations further) as well as still working on revising causality for nutrient fertilization.

Response: Thank you very much for your comments! In accordance with this comment as well as the recommendation from the Editor, we have systematically replaced the term "abundance" with "biomass" for consistency throughout the manuscript. We have also addressed your comment regarding wind data in a latter part of this document.

While the correlation between Chl-a and sGMW is compelling, the inference that meltwater acts primarily through nutrient fertilization, especially via iron, remains speculative. No direct nutrient data (e.g., Fe, NO₃⁻, PO₄³⁻, Si(OH)₄) are presented. The manuscript discusses literature findings from other regions (e.g., fjords, Andvord Bay), but these cannot be assumed to be representative of the broader WAP shelf over the 20-year period studied here. Similarly, potential contributions from upwelling, lateral advection, or benthic fluxes are not considered. While some of these are mentioned in the discussion, they are not integrated into the data analysis or explicitly tested. The revised manuscript improves clarity around this issue and now refers to nutrient input as a "likely" mechanism in some places. However, other sections still present this conclusion too strongly. For instance, the manuscript states that the strong positive correlation between sGMW and Chl-a points to nutrient enrichment from glacial meltwater. This causal language should be revised to make clear that the mechanism is hypothetical and not directly measured in this study.

Response: Thank you for your insightful comments. We have revised the manuscript to clarify our interpretations, carefully distinguishing our results and prior established studies from any inferred causal relationships. We acknowledge your important point regarding the absence of direct dissolved

iron (dFe) measurements in glacial meltwater from this study (Page 8, L.220 ["Although this study does not include direct measurements of dFe, earlier works studies demonstrate..."]). However, several previous studies (now explicitly referenced in our manuscript, Page 8, L.223) provide robust evidence supporting the relationship between glacial meltwater and iron enrichment in coastal marine environments:

- Hawkings et al. "Ice sheets as a significant source of highly reactive nanoparticulate iron to the oceans." *Nature communications* 5.1 (2014): 3929.
- Raiswell et al. "Contributions from glacially derived sediment to the global iron (oxyhydr) oxide cycle: Implications for iron delivery to the oceans." *Geochimica et Cosmochimica Acta* 70.11 (2006): 2765-2780.
- Raiswell "Iceberg-hosted nanoparticulate Fe in the Southern Ocean: Mineralogy, origin, dissolution kinetics and source of bioavailable Fe." *Deep Sea Research Part II: Topical Studies in Oceanography* 58.11-12 (2011): 1364-1375.
- Forsch et al. "Seasonal dispersal of fjord meltwaters as an important source of iron and manganese to coastal Antarctic phytoplankton." *Biogeosciences* 18.23 (2021): 6349-6375.

Collectively, these studies underpin our interpretation that glacial meltwater can enhance coastal iron concentrations. Additionally, we have ensured the manuscript explicitly acknowledges alternative fertilization mechanisms and discusses uncertainties appropriately.

One suggestion is for the authors to ensure clarity wherever they discuss correlations: differentiating clearly between what their data show (correlation between melt and Chl-a) and what they hypothesize as the cause (iron/stability), backed by prior studies. A sentence or two explicitly framing it as a "likely mechanism supported by [literature]" would guard against readers over-interpreting correlation as causation.

Response: Thanks for your comments! We have revised the manuscript to provide additional clarity regarding this point. Please see our revisions to your specific comments below.

Whenever authors talk about anything "significant" (or "not significant"), they need to specify to what degree (e.g. with p-values). Please review this throughout the text or change your wording accordingly.

Response: Thanks for this comment. We have revised all instances where the term "significant" does not refer to a statistical analysis.

While Chlorophyll-a concentration can more acceptably be used (upon limitations acknowledgement) as a proxy for phytoplankton biomass, caution is warranted when interpreting Chl-a as a direct proxy for cell abundance. In general, Chl-a is a poor indicator of abundance due to physiological, taxonomic, and environmental variability that can bring the same Chl-a concentrations to represent completely different abundances. This is widely acknowledged in the literature. I invite the authors to reflect on this and modify the entire text accordingly, including the title, by pointing to Chl-a as a proxy for biomass rather than abundance.

Response: Thanks for this comment. As stated in the prior response, we have replaced the term "abundance" with "biomass" throughout the manuscript and amended the Methods section to reflect these important nuances in the chl-a measurement.

In the Discussion or Conclusions, it might be valuable to include one sentence about phytoplankton community composition as a future consideration. While the study focuses on total biomass, the authors have the expertise to note that the ecological impact of meltwater could also involve shifts toward certain taxa (for example, meltwater has been associated with smaller cryptophytes in some WAP fjords, whereas diatoms dominate in stratified but nutrient-rich conditions). A brief mention that the community structure and broader food-web implications were not directly assessed here and remain important topics for future research would acknowledge the "ecology" aspect. Such a statement would add a forward-looking perspective and reassure readers that the authors recognize the complexity beyond chlorophyll concentration alone.

Response: Thank you very much for this valuable suggestion. We fully agree that phytoplankton community composition and the associated food-web implications represent critical components of understanding the ecological impacts of glacial meltwater in the WAP. While our current analysis focuses primarily on chl-a as a proxy for total phytoplankton biomass, we acknowledge that meltwater dynamics are likely to influence phytoplankton

communities in complex and potentially region-specific ways, as previous studies suggest (e.g., from WAP fjords such as Andvord Bay).

Following your suggestion, we have included a forward-looking statement in the Discussion (L.249--264) highlighting community composition as an important area for future research. We remain cautious about making definitive statements regarding the exact taxa favored by glacial meltwater inputs, given that existing results primarily come from localized fjord studies and the extent of their broader applicability along the Antarctic coast remains unclear. Additionally, current evidence suggests that glacial meltwater might facilitate seasonal succession among different phytoplankton groups rather than consistently favoring specific taxa. We hope this clarification captures both your helpful suggestions and our necessary caution in interpreting community-level impacts.

The authors should do a careful read-through for minor language issues. For instance, in one sentence the phrase “For an example, in Andvord Bay...” is used – this should be “For example, in Andvord Bay...”. Such small grammatical errors are few, but correcting them will improve readability.

Response: We have corrected these issues accordingly. Please see our responses and revisions according to your specific comments below.

Below, I provide my specific comments.

Response: Thank you again for your very detailed and thoughtful comments! Please see our responses below to address each of your comments.

Specific comments

I believe that a title that reflects your actual work is:

“Impact of Glacial Meltwater on Phytoplankton Chl-a along the Western Antarctic Peninsula”

or at maximum:

“Impact of Glacial Meltwater on Phytoplankton biomass along the Western Antarctic Peninsula”.

Response: Thank you for your comments! According to this comment as well as the Editor’s suggested title (which recommended the term “biomass”), we have replaced the term “abundance” throughout the manuscript and now use the term “biomass” instead.

Line 8. “However” does not fit here. The two sentences can be simply linked.

Response: After deleting the word “however,” there is no linkage between the two phrases. We have replaced the word with “but” to join the sentences. The text has been revised accordingly.

I still find the use of “ecology” inappropriate in several instances. Some suggestions here and further:

Line 11. “ecology” can be replaced by “biomass”.

Response: The text has been revised accordingly.

Line 15-17: “with an additional potential contribution from surface ocean stabilization, although the latter effect can be moderated by wind-driven mixing and sea-ice variability”. I suggest changing to e.g., “with an additional potential contribution from surface ocean stabilization due to the presence of sea ice”.

Response: The text has been revised accordingly.

Line 17-19: This sentence is still unclear and not entirely true. I’d suggest replacing it with e.g. “Achievable phytoplankton biomass depends on the interplay between light and nutrient limitation. Our results indicate that....”

Response: The manuscript has been revised accordingly but retained some of the original information that we were trying to convey in the abstract. The sentence now reads: “While high

phytoplankton biomass typically follows prolonged winter sea-ice seasons and depends on the interplay between light and nutrient limitation, our results indicate that ...”

Line 30-31: Why is only the 20th-century warming reported? Could you please provide more updated information, including the past 25 years? Especially cause the data later presented are within this latter range.

Response: Updated information has been added to include information from the last 25 years (L.32 Page 1, “This warming trend persists into the twenty-first century ...”).

Line 38: Please be sure this reference points to “abundance”, not something else.

Response: The term “abundance” has been replaced with “biomass;” it’s also consistent throughout the entire text (L.40, page 1)

Line 38: Replace “diatoms “ with “algae”.

Response: The text has been revised accordingly.

Line 39,40: “role” twice, consider replacing one of them.

Response: The text has been revised accordingly.

Line 42-43: Replace “diatom population that is released into the water column during ice melt” with “food source”.
Antarctic krill can access sea-ice brines

Response: The text has been revised accordingly. It now reads: “These algae can play enable ‘seeding’ by providing an early-season diatom population that is released into the water column during ice melt and serving as a food source for Antarctic kill.”

Line 48-49: “beyond the growing season” should be removed.

Response: The text has been revised accordingly.

Line 53: You should add here something about which kind of “taxonomic shifts”. Please revise.

Response: The text has been revised accordingly in L.53-54, page 2. “(e.g., “shifts between...”)

Line 68: “locales”?

Response: I believe this is the appropriate terminology because we are discussing “*n. a place where something happens or is set, or that has particular events associated with it*” – in this case, where sediment-laden glacial meltwater is released into coastal oceans.

Line 72: Remove “an” between “for” and “example”.

Response: The text has been revised accordingly.

Line 74-75: You should explain in which way the phytoplankton community and biomass were found to be affected.
Please, revise.

Response: The text has been revised accordingly to indicate how glacial meltwater impacts phytoplankton community and biomass in a prior study site.

Line 75-76: Since you made a distinction between dFe in spring and fall, you should explain to what season your pFe concentration corresponds. Please, revise.

Response: The reported range for pFe spans both seasons. The text has been revised accordingly.

Line 76: Replace “particular” with “particulate”.

Response: The text has been revised accordingly.

Line 92-93: should be removed as results are not presented yet.

Response: Thank you for your suggestion. In accordance with the publication’s style and formatting guide (commsj-phys-style-formatting-guide-accept.pdf), the final paragraph of the Introduction should be “a brief summary of the major results and conclusions.”

Line 107-108. I still believe that adding data on winds would help you solve the puzzle of the MLD. I haven’t asked you earlier to modify the current figure 2 but to add information on winds, which could be done in another panel and in relation to the other variables you have studied. Wind reanalysis are widely accessible and available and could be easily extracted for the study area and time period of interest.

Response: Thank you very much for highlighting the potential value of incorporating wind data in our analysis of MLD. We fully agree that winds significantly influence MLD variability in this region. However, the primary objective of this manuscript is to characterize the broad, long-term relationship specifically between sea-surface glacial meltwater and phytoplankton biomass at seasonal and interannual scales. Including wind data, which is inherently ephemeral (hourly or shorter variability), would substantially broaden the manuscript’s scope and complexity. Averaging or otherwise summarizing wind data to match our monthly-to-annual time scales could introduce additional analytical uncertainties and complications, potentially requiring significant new analysis to address temporal integration issues.

To address your valid concern, we have explicitly acknowledged (L.137-165) the importance of wind-driven mixing events in moderating the stabilizing effect of sGMW. We have also strengthened our discussion by citing detailed prior studies that specifically examine wind effects on MLD along the WAP. We believe this revision addresses your concern while preserving the focused analytical framework of our current study.

Line 110: change “abundance” to “biomass”.

Response: The text has been revised accordingly.

Line 110-111: “especially during broad-scale events captured in climatological and multi-decadal time series”, this statement is unclear, please revise.

Response: The text has been revised accordingly. It now reads: “especially when examining regional-scale processes over climatological and multi-decadal periods.”

Line 110, 116, 123: change “abundance” to “biomass”.

Response: The text has been revised accordingly.

Line 113: change “variability” to “biomass”.

Response: The text has been revised accordingly.

Line 117-118: I don’t understand why you report on other polar regions, and the next sentence is back to the WAP. Please, explain/revise.

Response: You previously recommended “... as the statement in lines 70-72 is not universally applicable” so we added this sentence to indicate the statement does not apply to all polar regions. We agree with your assessment and have deleted this sentence accordingly.

Line 123-125: This sentence contains a statement that should be explained or referenced.

Response: This sentence is a summary and conclusion regarding Figure 2 before transitioning to a discussion regarding information presented in this figure. We have added references to Figure 2 to clarify this point.

Caption Fig.2: Change “abundance” to “Chl-a”.

Response: The text has been revised accordingly.

Line 132: Please be sure this reference points to “abundance”, not something else.

Response: Yes, that is the case. Thanks for pointing this out.

Line 132: It is unclear what “this way” refers to. Please, revise.

Response: It has been deleted for clarity.

Line 137, 140, 144: change “abundance” to “biomass”.

Response: The text has been revised accordingly.

Line 151-152: This statement should be verified with actual data or referenced.

Response: The text has been revised accordingly, which included additional references.

Line 167: I believe a “over” is missing between “dominate” and “the”.

Response: The text has been revised accordingly.

Line 198: I suggest replacing “largely” with “likely”.

Response: The text has been revised accordingly.

Line 204: This “shift in climate conditions” should be further explained. Please, revise.

Response: The text has been revised accordingly to indicate the shift is characterized by changes in regional temperature, atmospheric circulation, and sea-ice patterns.

Line 206: cannot be called “trends” during such a short time period, please remove it

Response: The text has been revised accordingly.

Line 214: What are the “upper-bound uncertainty estimates”? Please revise/explain.

Response: Thank you for pointing this out. The term “upper-bound” has been deleted to provide clarity.

Line 223: “In contrast” does not fit here, I'd remove it.

Response: The text has been revised accordingly.

Line 227: I am unsure how “remote sensing efforts” can help “dFe data”? Please, explain/revise.

Response: Thanks for your inquiry. We are currently developing a hyperspectral ocean color based algorithm for detecting sea surface dFe.

Line 229: Please, remove “ecological”. This should be written as “the impact of sea-ice loss on surface Chl-a concentrations...”

Response: The text has been revised accordingly.

Line 234-235: I suggest replacing with “ Sea ice can seed and initiate ...”

Response: The text has been revised accordingly.

Line 280: Remove “abundance/”.

Response: The text has been revised accordingly.

Line 287, 289: Replace “abundance” with “biomass”.

Response: The text has been revised accordingly.

Line 309-318: The description of how the sGMW is derived from satellite imagery is especially crucial. I would encourage the authors to ensure that enough of that method is summarized here (or in the supplemental material) so that readers of this paper can understand it without needing to dig deeply into the other paper.

Response: Thanks for this comment. There’s a separate publication that describes this method in detail: Pan et al. (2023). Remote sensing of sea surface glacial meltwater on the Antarctic Peninsula shelf. *Frontiers in Marine Science*, 10, p.1209159, and it is also references in the Methods section.

Line 369-371: I’d move these lines within Fig. 1’s caption.

Response: Thanks for this comment. It is probably more suitable to leave these lines here because they describe methodological details rather than focusing on the information presented in the figure. Moreover, these lines refer to the method of remotely deriving sGWM fraction which sequentially fits in the Data Description section; moving these lines to Fig. 1’s caption (near the beginning of the manuscript) might introduce confusion.